# The chaperonin CCT8 controls proteostasis essential for T cell maturation, selection, and function

Bergithe E. Oftedal [1,2,11], Stefano Maio [1,11], Adam E. Handel [1], Madeleine P. J. White[3], Duncan Howie[4], Simon Davis [5], Nicolas Prevot[1], Ioanna A. Rota[1], Mary E. Deadman[1], Benedikt M. Kessler [5], Roman Fischer [5], Nikolaus S. Trede[6], Erdinc Sezgin [7,10], Rick M. Maizels[3] & Georg A. Holländer [1,8,9 ✉]

T cells rely for their development and function on the correct folding and turnover of proteins generated in response to a broad range of molecular cues. In the absence of the eukaryotic type II chaperonin complex, CCT, T cell activation induced changes in the proteome are compromised including the formation of nuclear actin filaments and the formation of a normal cell stress response. Consequently, thymocyte maturation and selection, and T cell homeostatic maintenance and receptor-mediated activation are severely impaired. In the absence of CCT-controlled protein folding, Th2 polarization diverges from normal differentiation with paradoxical continued IFN-γ expression. As a result, CCT-deficient T cells fail to generate an efficient immune protection against helminths as they are unable to sustain a coordinated recruitment of the innate and adaptive immune systems. These findings thus demonstrate that normal T cell biology is critically dependent on CCT-controlled proteostasis and that its absence is incompatible with protective immunity.

[1] Developmental Immunology, MRC Weatherall Institute of Molecular Medicine, University of Oxford, Oxford OX3 9DS, UK. [2] Department of Clinical Science, University of Bergen, Bergen, Norway, K.G. Jebsen Center for Autoimmune Disorders, Bergen, Norway. [3] Wellcome Centre for Integrative Parasitology, Institute of Infection, Immunity and Inflammation, University of Glasgow, Glasgow G12 8TA, UK. [4] Sir William Dunn School of Pathology, University of Oxford, Oxford OX1 3RE, UK. [5] Target Discovery Institute, Nuffield Department of Medicine, University of Oxford, Oxford OX3 7FZ, UK. [6] Huntsman Cancer Institute, University of Utah, Salt Lake City, UT, USA. [7] MRC Human Immunology Unit, MRC Weatherall Institute of Molecular Medicine, University of Oxford, Oxford OX3 9DS, UK. [8] Paediatric Immunology, Department of Biomedicine, University of Basel, Basel, Switzerland. [9] Department of Biosystems Science and Engineering, ETH Zurich, Basel, Switzerland. [10] Present address: Science for Life Laboratory, Department of Women's and Children's Health, Karolinska Institutet, Solna, Sweden. [11] These authors contributed equally: Bergithe E Oftedal, Stefano Maio. ✉email: georg.hollander@paediatrics.ox.ac.uk

T cells are indispensable co-ordinators of the adaptive immune response and embrace different effector functions dependent on the context in which they recognize their cognate antigen[1]. The competence of T cells to respond adequately to antigenic challenges is inextricably linked to de novo protein expression and a change in the cell's protein homeostasis. Under optimal conditions, this process relies on a complex network of interconnected, dynamic systems that control protein biosynthesis, folding, translocation, assembly, disassembly and clearance[2]. Cell stress and other physiological demands on the cell's proteome can however result in challenges where nascent and metastable proteins are misfolded and, as a result, aggregate-entrapped polypeptides are formed. These conformational changes challenge, as toxic intermediates, a cell's functions and may impair its survival. Thus, protein quality control and the maintenance of proteostasis are essential for almost all biological processes. This is in part accomplished by a machinery of chaperones that catalytically resolve misfolded proteins from adopting a state of amorphous aggregates but assist them in assuming a native conformation[3,4].

Chaperones are operationally defined by their capacity to interact transitorily with other proteins in assisting de novo folding of nascent proteins, refolding of stress-denatured proteins, oligomeric assembly, protein trafficking and proteolytic degradation[5]. Chaperones that partake in de novo folding or refolding promote these conformational changes through the recognition of hydrophobic amino acid side chains of non-native polypeptides and proteins. Eukaryotic type II chaperones, aka chaperonins, are large (~800−900 kDa), cytoplasmic, protein complexes (designated CCT, chaperonin containing tailless complex polypeptide 1, TCP-1 or TCP-1-ring complex, TRiC) that are built as hetero-octameric cylinders formed from two stacked doughnut-like rings[6]. Each of the rings is composed of homologous, yet distinct 60 kDa subunits ($\alpha$, $\beta$, $\gamma$, $\delta$, $\epsilon$, $\zeta$, $\eta$ and $\theta$)[7]; CCTs recognize, bind and globally enclose protein substrates of up to ~60 kDa to allow their folding over several cycles. Multiple substrates are recognized by CCTs whereby each of the complex's subunits may identify different polar and hydrophobic motifs[8]. High-affinity substrates are evicted from CCTs in an ATP-dependent fashion as they act as competitive inhibitors of the complex's catalytic reaction of unfolding proteins[9].

As many as 10% of newly synthesized proteins are assisted by CCT to adopt a correct conformation[10], including key regulators of cell growth and differentiation, and components of the cytoskeleton[11,12]. Actins and tubulins have been identified as two of the major folding substrates of CCT[13]. Adequate production of effector cytokines by T cells has been related to rapid actin polymerization and the generation of a dynamic filament network in the nucleus of CD4[+] T cells[14]. Moreover, signal transduction, cytoskeletal synthesis and remodelling, immune synapse formation, macromolecular transport and cell division require molecules that depend on CCT function[15,16].

To dissect the precise role of CCT in T cell development and function, we generated mice that lack the expression of a single subunit, CCT$\theta$ (aka CCT8), in immature thymocytes and their progeny. This particular subunit was chosen as previous investigations had identified CCT8 to be upregulated in activated and polarized T cells[17] and to be a key regulator in assembling the TriC complex[18]. We tested the proficiency of these cells to develop normally, be selected within the thymus, and respond to antigens as part of an adaptive immune response. Our results show that CCT8 is largely, albeit not entirely, dispensable for thymocyte differentiation and selection but essential for mature T cells to respond adequately to antigenic stimuli.

## Results

### Thymocyte development and peripheral T cell differentiation depend on CCT8 expression

The subunit 8 of type II chaperones, CCT8, was detected throughout thymocyte development but was most prominently found in immature cells with a double negative (DN, i.e. CD4−CD8−) phenotype (Fig. 1a). CD4-Cre::CCT8$^{fl/fl}$ mice (designated CCT8$^{T-/-}$; Supplementary Fig. 1 and Supplementary Data 1 and 2) have a loss of CCT8 expression targeted to double positive (DP i.e. CD4$^+$ and CD8$^+$) thymocytes and their progeny (Supplementary Fig. 1). The total thymus cellularity of these mice was comparable to that of Cre-negative littermates (designated CCT8$^{T+/+}$; Fig. 1b). As expected, the frequency of DN and immature single positive (iSPCD8) thymocytes remained unaffected as the deletion of CCT8 occurs after these maturational stages (Fig. 1c and Supplementary Fig. 1). However, the frequency of DP thymocytes was mildly increased and their progression to a post-signalling stage (TCRβ$^{hi}$CD69$^-$) was impaired (Fig. 1d), which correlated with a higher number of thymocytes that had not received a sufficiently strong survival signal (Fig. 1e). In parallel, the extent of negative selection during the first selection wave, as identified by the co-expression of Helios and PD1 on Foxp3−CCR7−TCR$^+$ DP or CD4 thymocytes (aka wave 1a and b, respectively), was reduced in CCT8$^{T-/-}$ mice (Fig. 1f, g). The following wave, which takes place in the medulla and is characterized by Helios expression on Foxp3− SPCD4 thymocytes, was reduced in a first (CD24$^+$) but not a second phase (CD24$^-$); Fig. 1h, i. Finally, fewer phenotypically and functionally mature single positive CD4$^+$ (SP4) and SPCD8 thymocytes were detected at a late stage of their development (Fig. 1c, j, k and Supplementary Fig. 1) and the frequencies of thymic and recirculating regulatory T cells (T$_{reg}$) were likewise reduced (Fig. 1l, m). Hence, the targeted loss of CCT8 expression in thymocytes impaired their selection and reduced the frequency of post-selection, mature effector and regulatory T cells.

The total splenic cellularity of CCT8$^{T-/-}$ mice was normal, although drastically fewer naive T cells were detected (Fig. 2a, b and Supplementary Data 1 and 2) and the CD4 and CD8 lineages were differentially affected (Fig. 2c). The frequencies of CD4 and CD8 T cells with a memory phenotype was increased (Fig. 2d), likely reflecting homeostatic expansion as a consequence of low T cellularity. Correspondingly, the frequency of peripheral T$_{reg}$ cells (CD25$^+$FoxP3$^+$) remained unaffected in CCT8$^{T-/-}$ mice but the subpopulation of highly suppressive CD103$^+$ ICOS$^+$ T$_{reg}$ was several fold increased in line with the extent of lymphopenia (Fig. 2e, f)[19]. Moreover, the frequency of CD4$^+$ memory T cells with an anergic phenotype (CD73$^+$FR4$^+$) was reduced in CCT8$^{T-/-}$ mice (Fig. 2g). Collectively and contrary to the relatively minor decrease in thymic SP cells, the loss of CCT8 expression correlated with a severe reduction in peripheral T cells implying a functional impairment of these cells.

### Loss of CCT8 impairs the formation of nuclear actin filaments

To assess the proliferative capacity of T cells deficient or proficient for the expression of CCT8, naive CD4$^+$ T cells were labelled with the membrane dye cell trace violet, activated (using cross-linking with anti-CD3 and anti-CD28 antibodies) and cultured for a total of 96 h. Activated CCT8$^{T-/-}$ T cells displayed a drastically reduced expansion index when compared to controls and their survival was greatly reduced, which could not be enhanced by the addition of IL-2 or the anti-oxidant N-Acetyl-L-cysteine (Fig. 3a, Supplementary Fig. 2, Supplementary Table 1 and Supplementary Data 1 and 2)[20]. Because T cell activation is associated with significant de novo protein synthesis[21] and the engineered lack of CCT8 expression reduced the expression of all components of the CCT complex (Fig. 3b), we next quantified in

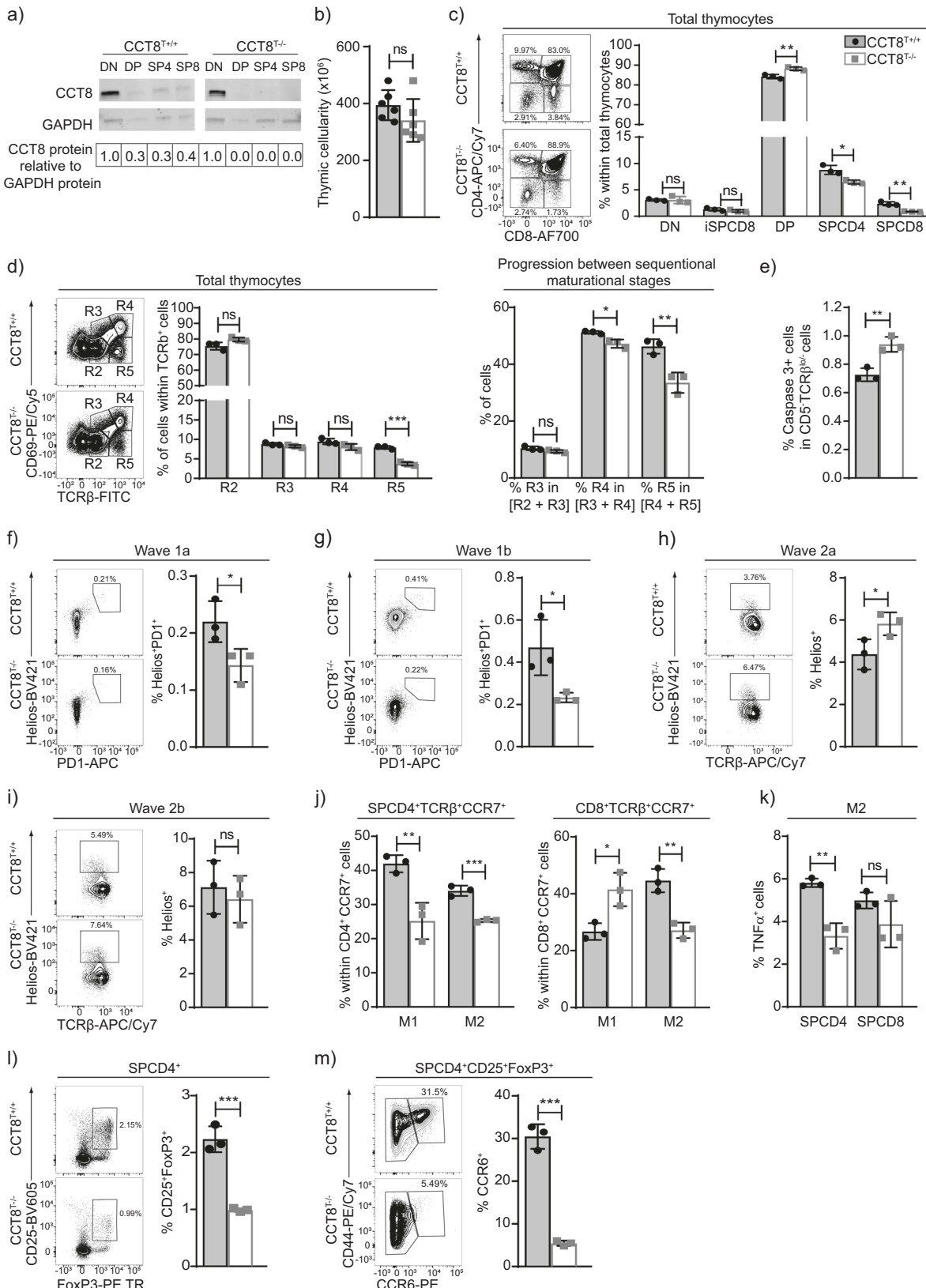

CCT8-deficient and -proficient T cells changes in tubulin and actin expression, as these two serve as folding substrates for the CCT complex[12,22]. Several tubulin isoforms were reduced in CCT8$^{T-/-}$ T cells, independent of the activation state, while the detection of the ubiquitously expressed β and γ actin isoforms was unaffected by a loss of normal CCT expression (Fig. 3c and

Supplementary Data 3). However, the formation of nuclear actin filaments could only be detected in a very small fraction of CCT8-deficient T cells (3.17 ± 1.15% versus 29.40 ± 10.5%) similar to the frequency of wild-type T cells in which Arp2/3 was pharmacologically inhibited and actin nucleation was prevented (Fig. 3d, e)[23]. Hence, the formation of nuclear actin filaments, an

**Fig. 1 The role of CCT8 in thymocytes. a** CCT8 protein detection by western blot. Values shown are relative to the detection in DN thymocytes. **b** Total thymic cellularity, and **c** thymocyte subpopulations in 6-week-old CCT8$^{T+/+}$ (grey bars) and CCT8$^{T-/-}$ mice (white bars) as defined by CD4 and CD8 cell surface expression on lineage-negative thymocytes. **d** Positive thymocyte selection and maturational stages (R2: TCRβ$^{low}$CD69$^-$; R3: TCRβ$^{low}$CD69$^+$, R4: TCRβ$^{hi}$CD69$^+$ and R5: TCRβ$^{hi}$CD69$^-$) and progression between sequential stages. **e** Non-signalled thymocytes (activated caspase 3$^-$ expression among CD5$^-$TCRβ$^{lo-}$ thymocytes). Negative thymocyte selection in the cortex. **f** Wave 1a (TCRβ$^{lo/hi}$ CCR7$^-$SPCD4$^+$SPCD8$^+$), and **g** wave 1b (TCRβ$^{hi}$ SPCD4$^+$ CCR7$^-$CD69$^+$CD24$^+$). Negative selection in the medulla. **h** Wave 2a (TCRβ$^{hi}$SPCD4$^+$CCR7$^+$CD69$^+$CD24$^+$), **i** and wave 2b (TCRβ$^{hi}$SPCD4$^+$CCR7$^+$ CD69$^-$CD24$^-$). **j** Late-stage maturation of single positive TCRβ$^{hi}$CCR7$^+$ thymocytes. Phenotypic analysis (M1: CD69$^+$MHC$^+$, M2: CD69$^-$MHCI$^+$). **k** TNFα production by M2 SPCD4 and SPCD8 cells. **l** Total thymic T$_{reg}$ cells (CD25$^+$Foxp3$^+$), and **m** recirculating T$_{reg}$ among thymic CD25$^+$Foxp3$^+$ cells. Left panels in (**c**, **d**, **f–i**, **l**, **m**) display representative contour plots of data shown in bar graphs (**b–h**). $^*p < 0.05$, $^{**}p < 0.01$, $^{***}p < 0.001$, $^{****}p < 0.0001$ (Student's $t$-test, (**b–m**), adjusted for multiple comparison panel (**j**)). Data shown in (**a**) are representative of two independent experiments. Data in bar graphs show mean ± SD representative two panels (**b–e** and **j–m**) and three (**f–i**) independent experiments, respectively, with three replicates each. See also Fig. S1.

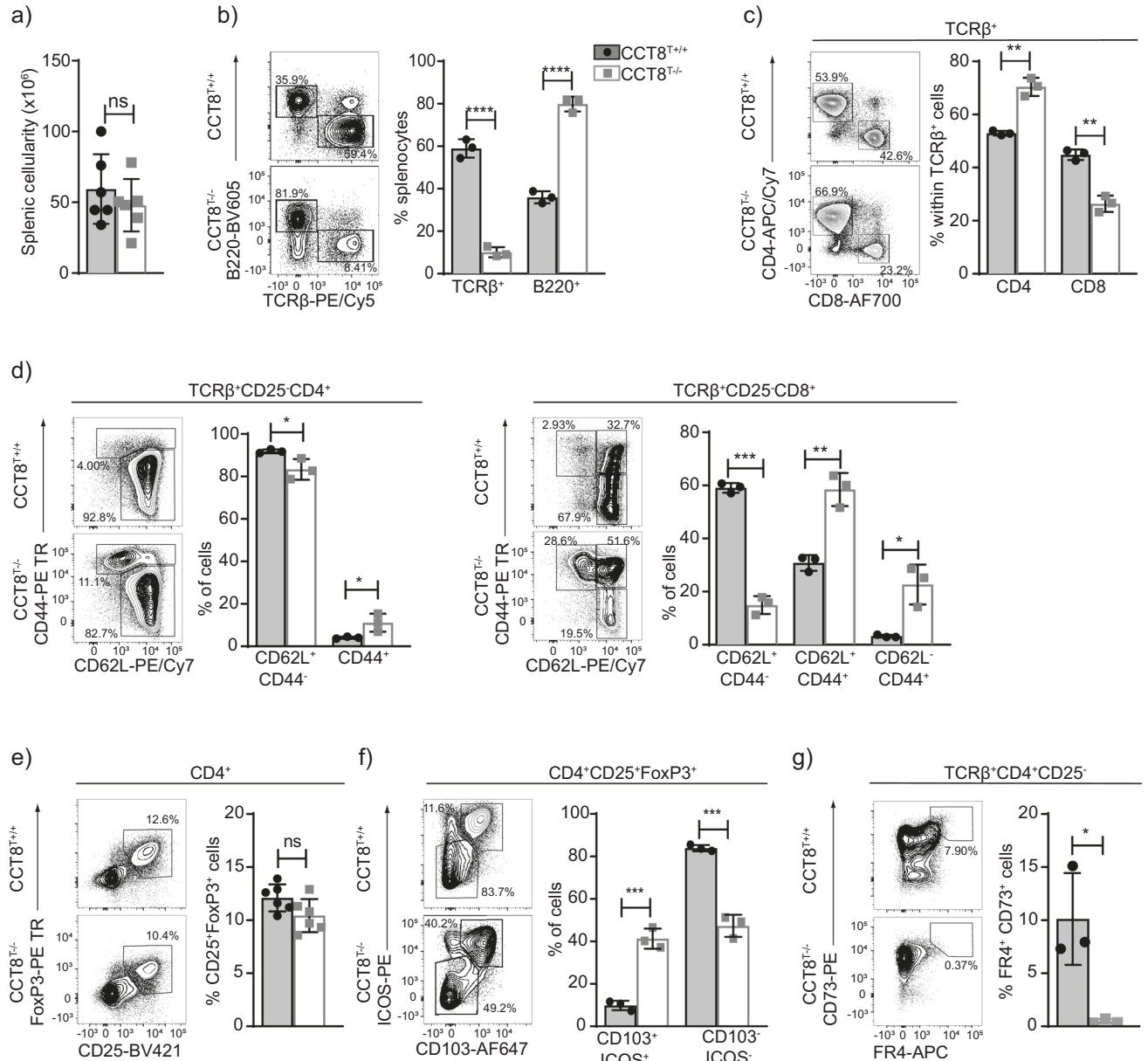

**Fig. 2 Peripheral T cell cellularity and phenotype in the absence of CCT8 expression.** Analysis of 4–6-week-old CCT8$^{T+/+}$ (grey bars) and CCT8$^{T-/-}$ mice (white bars). Gating strategies are shown for representative contour plots. **a** Total splenic cellularity. **b** Frequencies of splenic B and T cells. Frequencies (**c**) of total splenic CD4 and CD8 T cells, and (**d**) their naïve (CD62L$^+$CD44$^-$) and memory, effector memory (CD62L$^-$CD44$^+$) and central memory subpopulations (CD62L$^+$CD44$^+$), respectively. **e** The frequencies of total splenic T$_{reg}$ cells, and **f** T$_{reg}$ cell subpopulations defined by ICOS and CD103 cell surface expression. **g** Frequency of anergic CD4$^+$ T cells in lymph nodes. $^*p < 0.05$, $^{**}p < 0.01$, $^{***}p < 0.001$, $^{****}p < 0.0001$ (Student's $t$-test, (**a–g**)). Contour plots (**b–g**) are representative of data in bar graphs. Data shown in bar graphs represent mean ± SD values of a single experiment and are illustrative of two independent experiments with three replicates each.

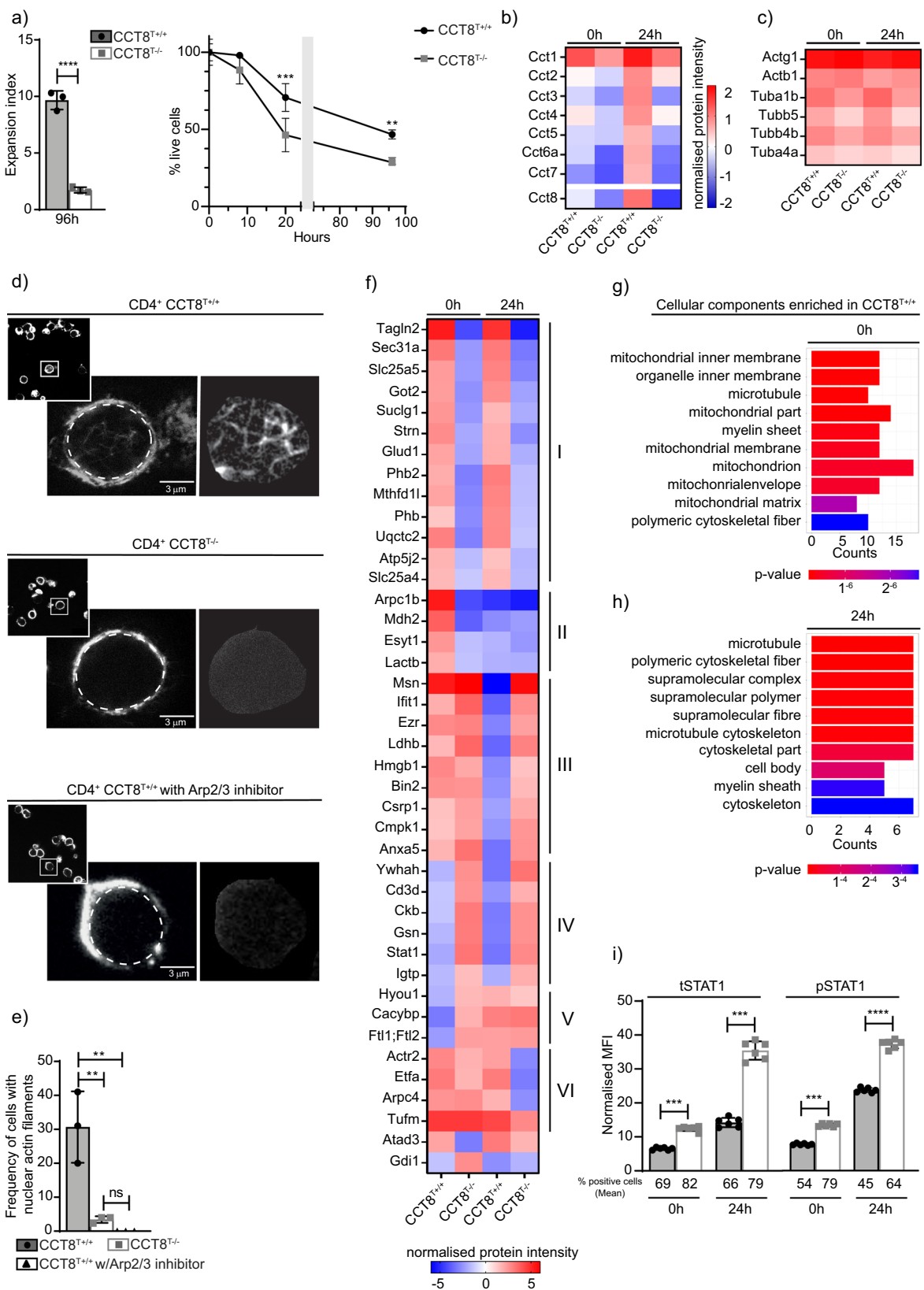

important requirement for T cell function[14], was dependent on an intact CCT complex.

**The lack of CCT8 compromises proteostasis in both resting and activated T cells.** Upon T cell activation, the lack of CCT8

also changed the expression of proteins other than individual CCT components and tubulin (Fig. 3f and Supplementary Data 3). At least six different patterns of protein changes were detected when comparing mutant and wild-type T cells before and 24 h after activation, including proteins that were minimally expressed in both resting and activated CCT8[T−/−] T cells (groups

**Fig. 3 Proteomic analysis of resting and activated CD4$^+$ T cells from CCT8T$^{+/+}$ and CCT8T$^{-/-}$ mice. a** T cells activated in vitro by CD3/CD28 cross-linking. Expansion index and normalized cell viability. **b** Expression of individual CCT subunits, and **c** the CCT substrates actin and tubulin measured by mass spectrometry. **d** Nuclear actin filaments in activated CD4$^+$ T cells in the presence of absence of CK-666, an inhibitor of Arp2/3. STED images with segmentation of the nuclear actin filaments (right panels), and **e** frequency of nuclear actin filaments detected in CCT8T$^{+/+}$ (white bars) and CCT8T$^{-/-}$ T cells (grey bars). The data display an average of 70 nuclei inspected in each of three independent experiments. **f** Differentially expressed proteins in resting (0 h) and activated (24 h) T cells. GO analysis of cellular components enriched in CCT8T$^{+/+}$ CD4$^+$ cells at 0 h (**g**) and 24 h (**h**) after stimulation. **i** The normalized MFI (MFI-positive population/MFI-negative population) of total STAT1 (tSTAT1) and phosohorylated STAT1 (pSTAT1) at 0 h and at 24 h upon activation in vitro by CD3/CD28 cross-linking. $^{**}p < 0.01$, $^{***}p < 0.001$, $^{****}p < 0.0001$. Data were calculated by Student's $t$-test (**a**, **e** and **i**) correcting for multiple comparisons (Holm-Sidak method; (**a**)) and ANOVA with post hoc test (**b**, **c** and **f**). Data shown in bar graphs represent mean ± SD values of a single experiment representative of two independent experiments with three replicates each, and results in heat maps and GO analyses are from two independent experiments with three replicates each. See also Fig. S2.

I + II), or that could be detected in unstimulated CCT8T$^{+/+}$ T cells but following activation did not change (groups III–V) or even a decrease in expression was detected (VI) (Fig. 3f). Many of the group I proteins were related to mitochondrial functions, for example, the ATP synthase membrane subunit f (Atp5j2, catalysing ATP synthesis[24]), glutamate dehydrogenase 1 (Glud1, catalysing the oxidative deamination of glutamate[25]) and prohibitin (Phb, controlling mitochondrial biogenesis and also cell-cycle progression, nuclear transcription and resistance to various apoptotic stimuli)[26,27]. CCT8T$^{+/+}$ CD4 T cells showed increased expression of mitochondrial-associated proteins and genes relative to CCT8T$^{-/-}$ CD4 T cells, which was in agreement with the cells' inadequate ability to increase protein synthesis and to adapt to metabolic demands (Fig. 3g, h and Supplementary Fig. 2). This deficiency in mitochondrial-related pathways was a consistent finding in both the proteome and transcriptome (Fig. 3g, h and Supplementary Fig. 2), and particularly focused on an altered expression of coenzyme Q10 metabolism and ATP synthesis. However, neither the biogenesis nor the membrane potential of mitochondria were impaired in activated T cells (Supplementary Fig. 2). We noticed in resting T cells a reduction in the concentration of Arpc1b (group II) and a lack of an upregulation of Arpc4 (group VI) in activated T cells, representing two of the five essential and non-interchangeable components of the Arp2/3 complex which promotes filamentous (F) actin branching (see Fig. 3d)[28]. The organizer proteins Moesin (Msn) and Ezrin (Ezr) (Group III) that link F-actin to the plasma membrane remained highly expressed in stimulated CCT8T$^{-/-}$ T cells thus impairing the remodelling of the cytoskeleton upon activation[29]. Furthermore, several proteins failed to be reduced in response to CD4$^+$ T cell activation (group III) including Ifit1, a protein expressed in response to interferons and the subsequent recruitment of STAT1 that also negatively regulates pro-inflammatory genes[30]. Moreover, flow cytometry confirmed that STAT1 and its activated form, phospho-STAT1 (pSTAT1) were increased both before and 24 h after mitogenic stimulation (Fig. 3i). RNA-Seq analysis showed differential gene expression and confirmed for activated CCT8T$^{-/-}$ T cells an enrichment of genes belonging to the IFN-γ pathway (GO:0034341; 8.8-fold change in mutant when compared to wild-type cells, adjusted $p$-value = 0.0002; Supplementary Fig. 2)[31].

**CCT8 is essential to avert T cell activation-induced cellular stress.** The accumulation of unfolded protein in the endoplasmic reticulum (ER) leads to cellular stress and, where unresolved, a loss of regular cell functions prompting apoptosis. Eukaryotic cells have developed an evolutionary well-conserved mechanism to clear unfolded proteins and to restore ER homeostasis, known as the unfolded protein response (UPR)[32]. The UPR comprises a tightly orchestrated collection of signalling events that are controlled by protein kinase RNA-like ER kinase (PERK), activating transcription factor 6 (ATF6) and inositol requiring protein-1α

(IRE-1α), which collectively sense ER stress and alleviate the accumulation of misfolded proteins, for example, via increasing the expression of ER chaperones[33] (Supplementary Fig. 3).

Naive CD4$^+$ and CD8$^+$ T cells from CCT8T$^{-/-}$ and CCT8T$^{+/+}$ mice, respectively, were activated for 48 h using anti-CD3 and anti-CD28 antibodies to detect UPR-related cell stress. Activated wild-type CD4$^+$ T cells exposed to Tunicamycin to induce ER stress were used as a positive control. Increased transcripts for *Perk*, *Atf6* and *Ire1-α*, were detected in activated CCT8T$^{-/-}$ CD4$^+$ T cells when compared to controls (Fig. 4a). In keeping with and consequent to an activated UPR, untreated CCT8T$^{-/-}$ CD4$^+$ T cells and treated CCT8T$^{+/+}$ CD4$^+$ T cells increased their transcripts for the chaperone *glucose-regulated protein* (*Grp*) *78*, a target gene of ATF6, and XBP-1 which is placed in the ER lumen and contributes there to ER homeostasis, protein folding and degradation[33,34]. Activation of IRE1-α also led to an upregulation of pro-apoptotic *Bim* and a decreased expression of the anti-apoptotic *Bcl2*, thus contributing to the impaired survival of activated CCT8T$^{-/-}$ CD4$^+$ T cells (Fig. 4a, Supplementary Data 1). In contrast, CCT8T$^{-/-}$ CD8$^+$ T cells displayed only a limited upregulation of *Perk* expression, whereas components of the other ER stress pathways were either unchanged or even diminished when compared to wild-type CD8$^+$ T cells. Taken together, activated CD4 + CCT8T$^{-/-}$ T cells displayed an extensive UPR as a result of impaired proteostasis, which appeared incompatible with normal T cell function.

**Th2 cell polarization and T cell metabolism are dependent on CCT8 expression.** Activated T cells undergo clonal expansion and differentiate, in the presence of additional molecular cues, into functionally distinct T subsets characterized by separate cytokine profiles and effector behaviours. Because mitochondrial and proteomic reprogramming parallel this differentiation, we next examined whether an inadequate mitochondrial response to metabolic demands, combined with an abnormal UPR (Supplementary Fig. 3), could impair peripheral T cell differentiation. Under Th1 polarizing conditions, the absence of CCT8 expression did not impair the viability of in vitro activated CD4 T cells but increased their frequency to express the signature cytokine IFN-γ (Fig. 4b and Supplemental Data 1 and 2). In contrast, both viability and IL-4 production were reduced in activated CD4 CCT8T$^{-/-}$ T cells polarized to adopt a Th2 phenotype (Fig. 4c). Paradoxically, the frequency of IFN-γ expressing CD4 CCT8T$^{-/-}$ T cells was increased by 7-fold under these conditions. Both of these findings are in agreement with an increased detection of STAT1 and pSTAT1 in CCT8T$^{-/-}$ T cells, independent of their activation status (Fig. 3e). Conditions favouring T$_{reg}$ differentiation not only caused a reduced viability in CCT8T$^{-/-}$ T cells, but also resulted in a reduced conversion of these cells to express FoxP3, whereas the vast majority of viable cells unexpectedly expressed IFN-γ (Fig. 4d). Driving CCT8T$^{-/-}$ T cells to a Th17 phenotype correlated with a dramatic loss in viability albeit the

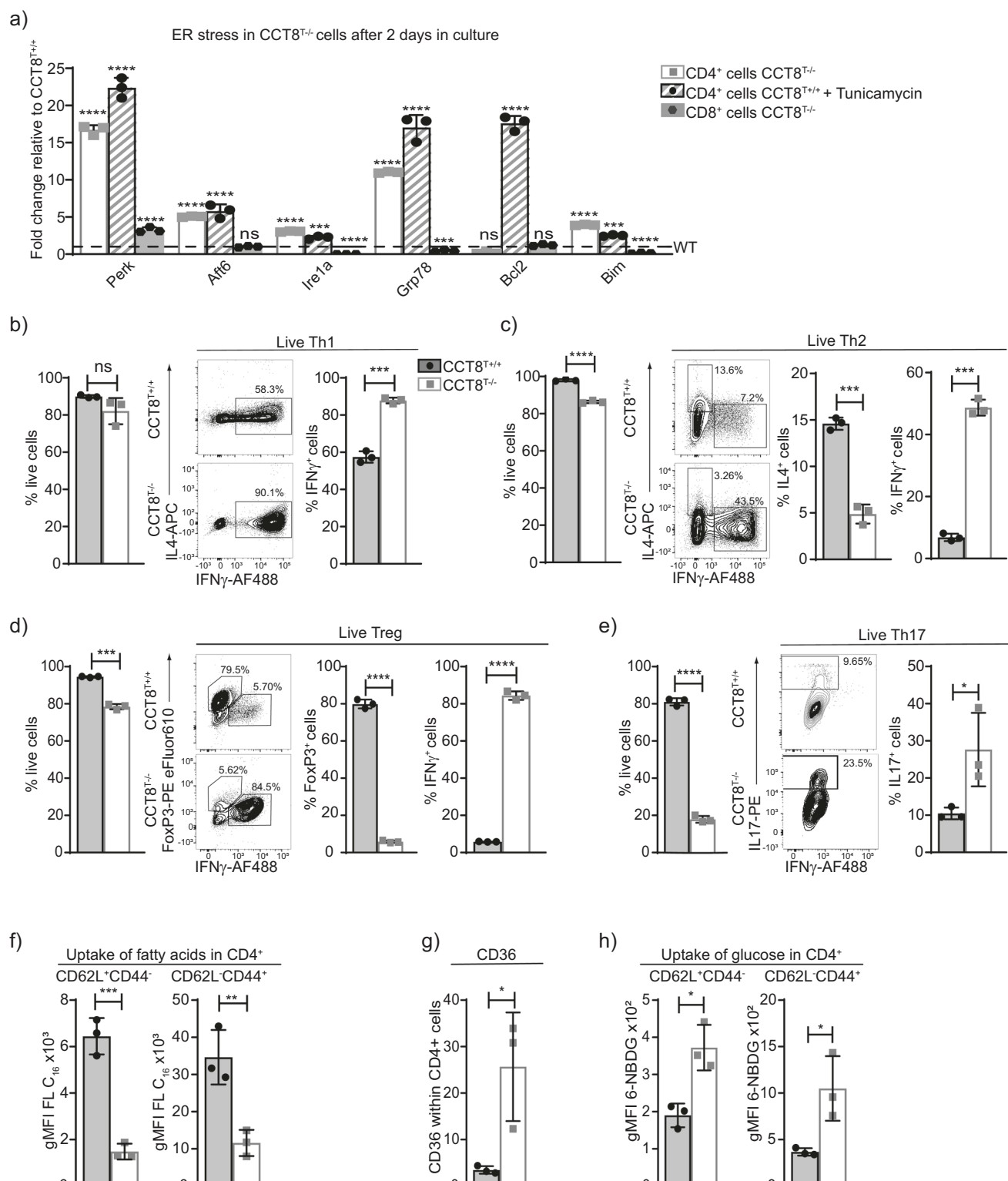

**Fig. 4 Peripheral T cell functions in the absence of CCT8 expression.** Analysis of naive CD4+ T cells from 4–6-week-old CCT8T+/+ (grey bars) and CCT8T−/− mice (white bars). **a** qPCR analysis of ER stress response elements in CD4+ and CD8+ T cells activated by CD3 and CD28 cross-linking and cultured in the presence or absence of Tunicamycin (CD4+ cells only); expression normalized to GPDH and displayed as 2−ΔΔCt CT values relative to values from CCT8T+/+ CD4+ and CD8+ T cells arbitrarily set 1. **b** In vitro differentiation of peripheral naive CD4+ T cells grown for 5 days under differentiating conditions: frequency of live cells (left) and cells (right) adopting: **b** Th1 polarization; **c** Th2 polarization; **d** Treg differentiation; and **e** Th17 differentiation. **f** Uptake of fatty acids in CD4+CD62L+CD44− (naive) and CD62L−CD44+ (memory) cells ex vivo activated by CD3 and CD28 cross-linking for 20 h. **g** Expression of long-chain fatty acid receptor CD36 on CD3/CD28-activated cells, and **h** uptake of glucose analogue 6-NBDG in CD4+CD62L+CD44− (naive) and CD62L−CD44+ (memory) cells. *p < 0.05, **p < 0.01, ***p < 0.001, ****p < 0.0001. Data were calculated by ANOVA correcting for multiple comparisons (Holm-Sidak method) (**a**) and Student's t-test (**b–h**). Bar graphs show mean ± SD and are representative of two independent experiments with three replicates each. See also Fig. S3.

frequency (but not the total cellularity) of living cells successfully polarized was increased (Fig. 4e). This reaction was CCT8$^{T-/-}$ T cell-intrinsic, as a comparable response was observed when CCT8-deficient and -proficient T cells were co-cultured during polarization (Supplementary Fig. 3). Thus, polarization of T cells was severely altered in the absence of CCT8, thus favouring these cells to adopt a Th1 phenotype.

As T cells transform from a quiescent to an activated state, the generation of energy from shared fuel inputs such as fatty acids and glucose are essential for the cells' growth, differentiation and survival[35,36]. We therefore tested whether an absence of CCT8 expression impaired the use of these two essential energy sources by CD4$^+$ T cells. Fatty acid uptake was significantly reduced in both activated naive and memory CD4$^+$ T cells lacking CCT8 (Fig. 4f). This result was notably independent of an increased cell surface expression of CD36, a glycoprotein that acts together with chaperones to translocate fatty acids to the cytoplasm (Fig. 4g). In contrast to wild-type controls, both naive and memory CCT8$^{T-/-}$ T cells displayed a higher glucose uptake (Fig. 4h) indicating a compensatory mechanism to be in play that secures, at least in part, the cells' energy expenditure. Indeed, basal respiration, spare respiratory capacity and extracellular acidification rate (ECAR), which measures glycolysis, remained globally normal despite a lack in CCT8 (Supplementary Fig. 3). Hence, the absence of CCT8 in T cells resulted in a change in energy usage which may further explain the impaired response to cell activation.

**CCT8 is essential for protective immunity against intestinal helminths.** Infection with the nematode *Heligmosomoides polygyrus* activates a strong Th2-type immune response[37]. Primary infections in susceptible strains such as C57BL/6 are normally non-resolving and clearance requires drug treatment with an anti-helmintic such as pyrantel embonate. In contrast, spontaneous clearance of secondary infections relies on a protective anti-helminthic immunity and is observed in mice previously exposed to *H. polygyrus* and subsequently drug treated[38]. We therefore investigated whether the Th2 polarization deficiency observed in CCT8$^{T-/-}$ mice would impair the expulsion of *H. polygyrus* after a short primary infection followed by a re-exposure to the parasite's larvae. In comparison to wild-type animals, a higher burden of worm eggs was detected in the faeces of infected CCT8$^{T-/-}$ mice, especially on re-exposure (Fig. 5a). This result correlated at the peak of primary and secondary infections with both fewer total cells and a reduced frequency of CD4$^+$ Th2 cells (as determined by IL-4 and Gata3 expression) in mesenteric lymph nodes and peritoneal lavage (Fig. 5b–d and Supplementary Data 1 and 2).

Resistance to *H. polygyrus* infections is closely related to the extent and speed of a pathogen-specific Th2 response, whereas susceptibility to these worms is linked to T$_{reg}$ activity and IFN-γ production[39,40]. We therefore analysed the frequency of T$_{reg}$ populations during primary and secondary *H. polygyrus* infections. In mesenteric lymph nodes, CCT8-deficient T$_{reg}$ cells were detected at higher frequencies during the primary and secondary pathogen challenges when compared to wild-type controls (Fig. 5e). However, fewer T$_{reg}$ were observed in the peritoneal lavage of infected CCT8$^{-/-}$ mice, possibly reflecting differences in their homing to or survival in situ, although these differences did not reach statistical significance (Fig. 5f). Hence, the insufficient priming and the subsequent absence of a protective immune response to a second *H. polygyrus* challenge correlated with the detected preference of CCT8$^{-/-}$ T cells to adopt a Th1 phenotype, their readiness to secrete IFN-γ and their reduced capacity to adopt a Th2 phenotype and paralleled a local expansion of T$_{reg}$ cells in mesenteric lymph nodes in response to priming to *H. polygyrus*.

The exposure to Th2 cytokines in the context of parasite infections promotes granulomatous reactions composed of alternatively activated macrophages (AAMacs). Identified by their elevated expression of the resistin-like molecule (RELM)-α[41], AAMacs were recovered from the peritoneal lavage of wild-type animals infected with *H. polygyrus* but were hardly detected at these sites in CCT8$^{T-/-}$ mice, even after re-exposure to the parasite (Fig. 5g). Similarly, the number of eosinophils in the peritoneal lavage was low in CCT8$^{T-/-}$ mice with a primary *H. polygyrus* infection and this paucity remained unchanged upon re-infection (Fig. 5h), thus correlating with the restricted capacity of CCT8$^{T-/-}$ mice to be polarized to a Th2 phenotype.

*H. polygyrus* infections activate B cells and elicit a strong humoral immune response supporting the clearance of the parasite[42]. We therefore also determined the absolute number and frequency of B cells in the peritoneal lavage of infected wild-type and CCT8$^{-/-}$ mice (Fig. 5i) and probed the secretion of *H. polygyrus*-specific serum IgG1. In comparison to infected controls, there was a higher frequency of B cells in the mesenteric lymph nodes of mutant mice but fewer in the peritoneal cavity during both primary and secondary infections (Fig. 5i and Supplementary Fig. 5). In parallel, CCT8$^{T-/-}$ mice failed to generate an antigen-specific IgG1 response to a secondary *H. polygyrus* infection (Fig. 5j) further highlighting the consequences of a limited Th2 response in vivo.

The type 2 cytokines IL-5 and IL-13 are also secreted early during a helminth infection by innate lymphocytes (ILC2) in response to the release of IL-33 and TSLP from mucosal epithelial sensor cells[43,44]. We therefore quantified these innate lymphocytes in mesenteric lymph nodes and in the peritoneal lavage of infected wild-type and CCT8$^{-/-}$ mice. During the initial infection, CCT8$^{T-/-}$ mice had a higher frequency of ILC2 at both anatomical sites but the absolute cellularity was comparable to that of control animals (Fig. 5b, k, l). Upon re-infection with *H. polygyrus*, the ILC2 frequency was comparable for CCT8$^{T-/-}$ and CCT8$^{T+/+}$ mice but the absolute cellularity was lower in mutant mice (Fig. 5k, l). This finding correlated with a lack of intestinal tuft cell expansion during secondary infection in mutant mice, suggesting the feed-forward loop, in which type 2 cytokines increases the numbers of these cells[44,45], was not supported by resident ILC2 cells indicating an adaptive T cell contribution is required which is absent in the CCT8$^{T-/-}$ mice (Supplementary Fig. 4). In addition, the scarcity of AAMacs, eosinophils and B cells in the peritoneal lavage of CCT8$^{-/-}$ mice demonstrated that the overall provision of type 2 cytokines in response to the nematode was inadequate to recruit an effective cellular and humoral response to *H. polygyrus*.

## Discussion

CCT captures and manipulates the folding of non-abundant intermediates from a range of substrates whereby its interaction occurs in a subunit-specific and geometry-dependent fashion[46]. We demonstrate here that a targeted loss of only the CCT8 protein compromised the function of the entire CCT complex and thus impaired the correct folding and consequent function of different substrates, including the cytoskeletal proteins actin and tubulin. Actin with its V-shaped molecular structure[47] has been identified as a prototype substrate that requires CCT's function for its efficient folding[46], a process involving the binding to CCT4 and either CCT2 or CCT5 but spares direct contacts with CCT8[46,48]. The biogenesis of actin is, however, unaffected by the depletion of CCT8, a finding in line with observations in *C. elegans* where actin levels remain normal despite an impaired CCT8

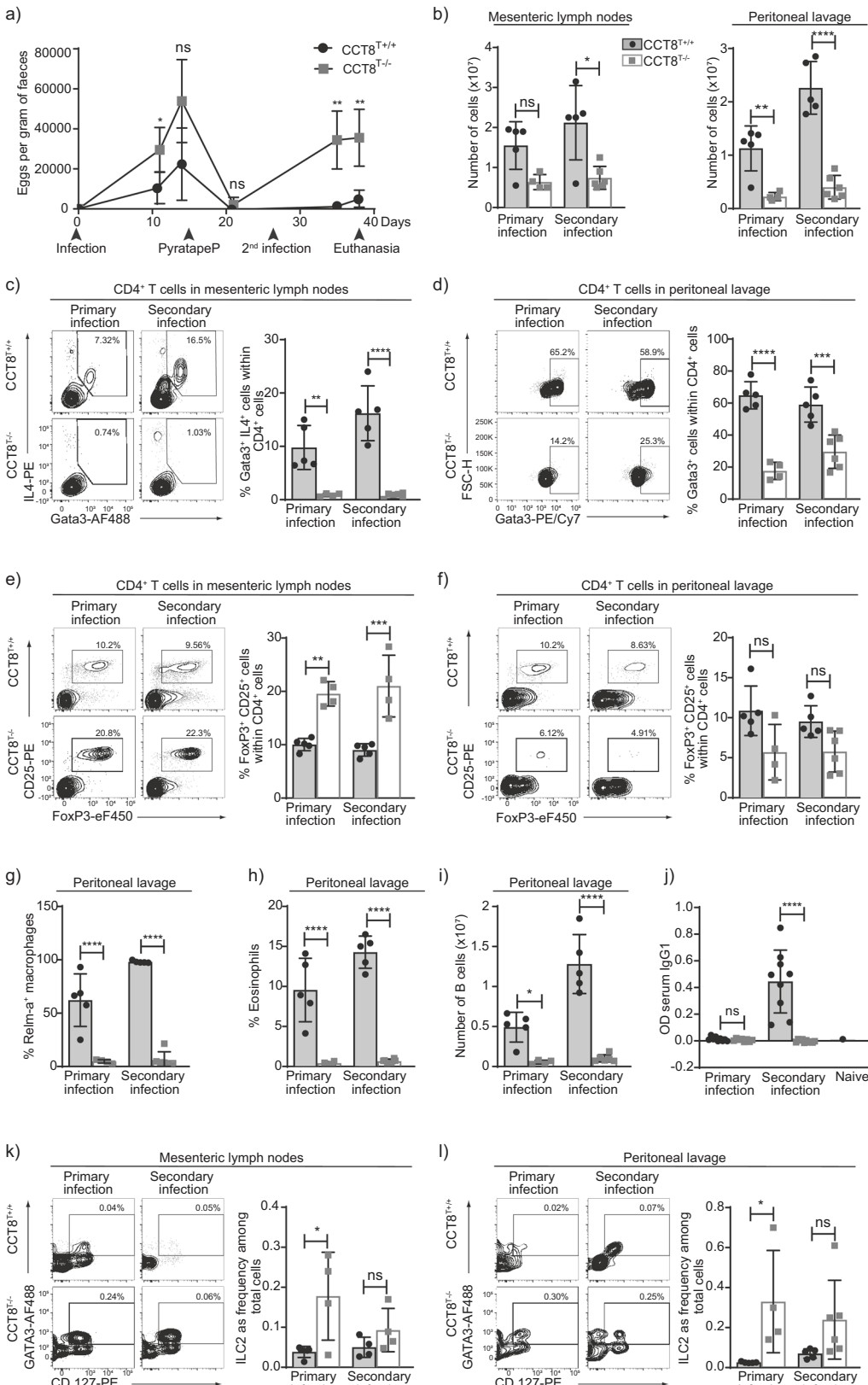

**Fig. 5 The response to primary and secondary *H. polygyrus* infection. a** Eggs per gram of faeces at indicated timepoints. Comparison of CCT8$^{T+/+}$ (black circles) and CCT8$^{T-/-}$ mice (grey squares). **b** Cellularity in mesenteric lymph nodes, and peritoneal lavage. **c** Th2 cell frequency in mesenteric lymph nodes, and **d** peritoneal lavage. **e** Treg frequencies in mesenteric lymph nodes, and **f** peritoneal lavage. Frequencies of (**g**) alternatively activated macrophage and (**h**) eosinophils in peritoneal lavage. **i** B cells in peritoneal lavage, and **j** *H. polygyrus*-specific serum IgG1. ILC2 frequency in mesenteric lymph nodes (**k**) and peritoneal lavage (**l**). **c–f** and **k, l** Display of contour plots with gating. *$p < 0.05$, **$p < 0.01$, ***$p < 0.001$, ****$p < 0.0001$. Data were compared by Student's *t*-test adjusted for multiple comparison (Holm-Sidak method). Bar graphs show mean ± SD and are representative of one of the two independent experiments with at least 4 samples per group.

function[22]. In this experimental system, aggregates including actin are not efficiently cleared from the cytoplasm causing the formation of aggregates and misfolded proteins that convey either a loss, a change or even a gain of function as a result of their toxicity.

Productive folding and processing of actin in T cells is an important prerequisite for the signalling-dependent changes in shape. Indeed, T cells adapt to different physiological conditions including, for example, the stress of the bloodstream flow, the migration to and residence in different tissues each with their bespoke microenvironments, the engagement with antigen-presenting cells and the formation of immunological synapses[49]. Actin polymerization in T cells occurs in response to TCR-mediated activation and rapidly creates, in the presence of the Arp2/3 complex and several formins, a dynamic network of cytoplasmic and nuclear filaments. These structures play a pleiotropic role in T cell activation as they control the promotion of conjugates, the activation and nuclear import of transcription factors, and, possibly, the internalization of the TCR[14]. Conversely, inhibiting the cytoplasmic and nuclear formation of actin filaments broadly impairs T cell biology[14].

Our results demonstrated that depleting CCT constrains the nuclear formation of actin filaments and results in a panoply of functional changes. For instance, TCR-mediated progression of CCT8-deficient thymocytes past several intrathymic checkpoints was impaired, including the cells' normal positive and negative selection. These findings align well with studies that have identified a role for UPR in thymocyte maturation and selection (reviewed in ref. [32]). Interestingly, the observed partial block in thymocyte maturation was relatively mild despite a complete absence of CCT8 at the DP and later stages of thymocyte maturation. In stark contrast, the periphery of CCT8$^{T-/-}$ mice is severely lymphopenic indicating peripheral T cells to be especially reliant on this complex for their maintenance and function. Indeed, both ER stress and UPR play an important role in the homeostatic maintenance of peripheral T cells and are disordered in the absence of CCT, especially CD4$^+$ T cells. In contrast, the UPR of CD8$^+$ T cells was minimally affected by the absence of CCT8 for reasons yet to be determined. The mass spectrometric analysis of both resting and activated CCT8$^{T-/-}$ T cells further confirms at the protein level that several molecules involved in mitochondrial functions, including Phb1 and Phb2, are expressed at lower concentrations in the absence of CCT8. Phb1 and Phb2 form a hetero-oligomeric complex that contributes, inter alia, to an UPR in mitochondria[50].

In addition to a compromised cell viability and a restricted clonal expansion, the absence of CCT in activated T cells also causes an increased ability to secrete IFN-γ and a limited efficiency to adopt a Th2 phenotype, even when exposed in vitro to ideal polarizing conditions. While T-bet favours the expression of IFN-γ the same factor represses the Th2 lineage commitment via a tyrosine-kinase-mediated interaction that disrupts the binding of GATA3 to its DNA binding motif[51]. We noted under non-polarizing condition an upregulation of total and activated STAT1 in CCT8$^{T-/-}$ cells both before and 24 h after activation of primary CD4 + T cells, whereas neither T-bet nor IL-12 receptor expression were detected. Though previously thought to play a role in STAT1 biosynthesis[52], CCT is apparently not critically required for the formation of STAT1. The upregulation of GATA3 and activation of signal transducer and activator of transcription 5 (STAT5) are two indispensable events for this differentiation process to occur.

The co-evolution of vertebrate hosts with helminths has resulted over the course of millions of years in a sophisticated immune defence that engages several effector mechanisms orchestrated by a robust Th2-type response. This reaction activates and mobilizes a suite of innate immune cells and local tissue responses[53]. During the worm's life cycle, infective larvae taken up by oral route invade the mucosa of the duodenum, cross its muscle layer and reach a space beneath the serosa from where adult parasites return 8 days later to access the intestinal lumen. Even though a primary infection does not resolve without drug treatment under the experimental conditions used here, a secondary exposure is completely cleared in wild-type mice when sufficient IL-4 (but not necessarily IL-13) is available to activate responsive epithelia[38,54] and AAMacs forming granulomatous cysts to encase the larvae that have penetrated the intestinal wall[38].

The activation and expansion of B cells secreting cytokines and immunoglobulins, in particular IgG1 in a T cell-dependent fashion, are known to be critical for effective anti-H. polygyrus immunity[55–57]. Resistance to H. polygyrus (re-) infection also results in the activation of group 2 innate lymphoid cells (ILC2)[58], which are required for the differentiation of conventional Th2 cells[58]. However, the limited capacity of CCT8$^{T-/-}$ mice to effectively polarize their CD4$^+$ T cells in vivo to a Th2 phenotype thwarts the clearance of H. polygyrus upon re-infection even in the presence of ILC2 cells. Total cellularity and the frequency of Th2-type T cells are significantly reduced in both mesenteric lymph nodes and the peritoneal lavage of infected CCT8$^{T-/-}$ mice when compared to wild-type animals. This reduction and its downstream cellular and molecular consequences—possibly in addition with the general lymphopenia of CCT8$^{T-/-}$ mice—explains mechanistically the animals' inability to expel H. polygyrus via a robust immune response that generates sufficient IL-4. Interestingly, ILC2 in both mesenteric lymph nodes and the peritoneal lavage increase in CCT8$^{T-/-}$ mice during the first worm exposure indicate that the initial recruitment of these cells is largely intact. However, these cells require a greatly expanded Th2 population, and subsequent cytokine secretion, to escalate a full-scale anti-helminth immune reaction including tuft cell proliferation and alternative activation of macrophages. Although ILC2 cells are an innate source of IL4, their reduced cellularity is insufficient to create an effective immune defence in CCT8$^{T-/-}$ mice to bypass their defective Th2 response. This and additional findings presented here indicate that Th2-type cells are indispensable for a protective immunity against H. polygyrus whilst positioned upstream of AAMac, eosinophils and B cells whose activation and expansion cannot be driven by ILC2 alone.

H. polygyrus infections also expand and activate the host's T$_{reg}$ population, especially early in an infection when these cells appear to outpace the proliferation of effector T cells and modulate the immune response[59]. An increased frequency of T$_{reg}$, possibly as a result of low-dose IL-4 exposure[60] was observed in the mesenteric lymph nodes of infected CCT8$^{T-/-}$ animals where these cells may further inhibit Th2 immunity[61]. However, the frequency of T$_{reg}$ was significantly reduced in the peritoneal lavage. The molecular reason for this compartmentalization of T$_{reg}$ remains so far unknown, but may reflect a deficiency in the cell's expression of CD103[59] and homing to the intestine. This could result from a lack in upregulating the gut-trophic chemokine receptor CCR9 and/or an irregular response to CCL25, a chemokine highly expressed in the epithelium of the inflamed small intestine and on postcapillary venules of the lamina propria.

In mouse strains susceptible to H. polygyrus infections, Th2 responses are counterbalanced by IFN-γ-producing CD4$^+$ and CD8$^+$ T cells[59]. The predilection of CCT8$^{T-/-}$ T cells to adopt a Th1 phenotype and secrete IFN-γ upon activation suppresses the formation of a type 2 immune response and inhibits cell proliferation, thus further contributing to a deviation away from protective immunity against H. polygyrus. It is tempting to

speculate that such a shift towards a Th1-type immune response due to an absence of regular CCT8 function may be harnessed therapeutically in non-infectious, Th2-driven pathologies (e.g. allergic diseases). However, the benefit of such an imbalance towards a Th1-weighted immune reaction requires further probing, for example, in the context of an anti-viral immune response where IFN-γ discloses both anti-viral and immuno-modulatory functions to combat the infection while minimizing collateral tissue damage[62].

Taken together, CCT-mediated protein folding is essential for normal T cell biology as their development, selection and function are severely impaired by a lack of normal proteostasis. In this study, we demonstrate for the first time that the loss of CCT function in T cells disturbs normal proteostasis and the dynamic formation of nuclear actin filaments, a prerequisite for normal cell-cycle progression and chromatin organization. As a consequence, the maintenance of a normal peripheral T cell pool is severely compromised which correlates with an inadequate mitochondrial response to metabolic demands and an abnormal UPR weakening the ability to cope with activation-induced cell stress. Finally, CCT function is required for T cells to be Th2 polarized and to stage a protective immune response against *H. polygyrus* via adequate feed-forward loops engaging multiple cellular effector mechanisms.

## Methods

**Materials availability**. Further information and requests for resources and reagents should be directed to and will be fulfilled by the corresponding author.

### Experimental model

*Mice*. The Cct8 KO mouse model (Cct8tm1a(KOMP) Wtsi, Project ID CSD45380i) was obtained from the KOMP Repository (www.komp.org) and generated by the Wellcome Trust Sanger Institute (WTSI). Targeting vectors used were generated by the Wellcome Trust Sanger Institute and the Children's Hospital Oakland Research Institute, as part of the Knockout Mouse Project (3U01HG004080), designed to delete exon 2 in the Cct8 gene. The Cd4-Cre transgenic mouse was originally developed at the University of Washington on C57BL/6 background[63]. Mice between 4 and 10 weeks of age were used for experiments. Animals were maintained under specific pathogen-free conditions and experiments were performed according to institutional and UK Home Office regulations.

### Methods details

*Flow cytometry, cell sorting and cell purification by magnetic-activated cell separation*. Cells from thymus, spleen and lymph nodes were isolated from wild-type and mutant mice and stained using combinations of the antibodies listed in Supplementary Data 4. Where needed, unmanipulated naive CD4$^+$ cells were enriched using magnetic separation (Miltenyi Biotec). Before staining, cells were resuspended at a concentration of $10 \times 10^6/100$ μL in PBS containing 2% FCS (Merck). Staining for cell surface antigens was performed for 20 min at 4 °C in the dark. Thymocytes analysed were lineage negative, i.e. lacked the expression of CD11b, CD11c, Gr1, CD19, CD49b, F4/80, NK1.1, GL3 and Ter119. The following panels were used: Thymocyte surface panel (DN), antibodies recognizing: CD8 AF700, CD4 APC/Cy7, TCRβ Pe, CD24 FITC, CD25 eflour450, CD44 Pe/eflour610, CD69 Pe/Cy5, CD5 Pe/Cy7. Negative selection panel: CD8a AF700, CD4 FITC, TCRβ APC-Cy7, CD24 PerCP-efluor710, PD1 APC, CD25 Biotin, NK1.1 biotin, CD69 PE-Cy5, CD5 PE-Cy7, Streptavidin BV605, CCR7 PE, Helios BV421, FOXP3 PE-efluor610. Positive selection panel and SM, M1 and M2, antibodies recognizing: CCR7 BV421, CD4 Pe/efluor610, CD8 AF700, CD25 BV605, CD69 Pe-Cy5, CD5, PerCP-Cy5.5, TCRβ FITC, MHC-1 Pe, CD24 APC, Sca-1 Bv510, CD44 Pe-Cy7 Qa2 Biotin, streptavidin APC-Cy7.Treg panel: CD4 APC-Cy7, CD8a AF700, CD3e PerCP-Cy5.5, CD25 efluor450, CD69 PE-Cy5, CD5 PE-Cy7, ICOS PE, CD103 AF647, CD4 FITC, FOXP3 Pe/efluor610, CCR6 APC. Memory/naive panel; CD4 APC-Cy7, CD8a AF700, B220 BV605, CD44 Pe/efluor610, CD62L PE-Cy7, TCRβ PE-Cy5, CD25 ef450, CD5 PerCP-Cy5.5, FR4 APC, CD73 PE. For total STAT1 and pSTAT1 staining, the cells were activated by Cell Stimulation Cocktail (eBioscience) for 1 h and incubated with pre-warmed Fixation Buffer (BioLegend) at 37 °C for 15 min before permeabilized by pre-chilled True-Phos Buffer (BioLegend) at −20 °C overnight (O/N). The cells were stained with antibodies recognizing pSTAT1 and STAT1 according to the manufacturer's instruction. T cell polarization panels; IFN-γ FITC, IL17 Pe, FoxP3 Pe/efluor610, Rorγ APC. Where needed, staining for CCR7 was performed for 30 min at 37 °C in a water bath, directly followed by the addition of other cell surface stains. For the identification of intracellular markers (Foxp3, IL-17, Helios, IFN-γ), Foxp3 Transcription

Factor Staining Buffer Set (eBioscience) was used according to manufacturer's instructions; intracellular stains were performed for 60 min at 4 °C in the dark. For the identification of IL-4, cells were fixed with 2% PFA and then permeabilized with 0.1% Saponin. Prior to cytokine staining, cells were incubated with Cell Stimulation Cocktail (eBioscience) for 2 h before adding monensin, and further incubated at 37 °C for a total of 5 h. Cell viability was measured using LIVE/DEAD Fixable Aqua Dead Cell Stain Kit (ThermoFisher Scientific) as per the manufacturer's instructions. After staining, the cells were acquired and sorted using a FACS Aria III (BD Biosciences) and analysed using FlowJo v10.

*Detection of TNF-α production*. Assay of TNF-α production was performed in accordance with[64]. Briefly, total thymocytes were activated with 1 μg/mL plate-bound anti-CD3 and soluble 1 μg/mL anti-CD28 in the presence of monensin (BioLegend). After 4 h, the cells were stained for surface markers (SM, M1, M2 panel), treated with Foxp3 Transcription Factor Staining Buffer Set (eBioscience) according to manufacturer's instructions before staining for intracellular TNF-α.

*Western blot*. 100,000 DN, DP, SP4 and SP8 cells were FACS sorted from 4 old CCT8$^{T+/+}$ ($n = 8$) and CCT8$^{T-/-}$ mice ($n = 8$) in 50,000 cell batches and stored as dry cell pellets at −80 °C until enough cells were collected. The pellets were lysed in 11.25 μL lysis buffer containing 25 mM Tris-HCl pH8 (Merck), 50 mM NaCl (Merck), 1% NP-40 (Merck), 1 m DDT (Merck), 10% glycerol (Merck) and 0.2 mM PMSF (Merck) in deionized water. After 15 min on ice, the samples were centrifuged at 15,000g for 10 min at 4 °C; 11.25 μL of each supernatant was added to 3.75 μL sample buffer containing 250 mM Tris-HCl pH7 (Merck), 10% SDS (Merck), 35% glycerol (Merck) and 0.05% Bromophenol blue (Merck), and heated at 97 °C for 10 min before gel electrophoresis. Proteins were separated on a BisTris 4–10% Gradient gel (Invitrogen) and the samples run alongside a protein standard (Biorad), before transferred to a polyvinylidene difluoride membrane (Biorad). Antibody recognizing CCT8 was added at a 1:1000 dilution, with IRDye 680RD goat anti-rabbit antibody (Licor) diluted in 1:10,000 as secondary antibody, before imaging on an Odyssey imaging system (Licor). As a control for protein loading, the membrane was re-stained for GAPDH (AbCam). The blot is shown is Supplementary Data 5.

*Primary cell culture, activation, proliferation and polarization*. Naive CD4$^+$ T cells were labelled by Cell Trace Violet (ThermoFisher Scientific) according to the manufacturer's protocol and activated in vitro using a combination of plate-bound anti-CD3 (2 μg/mL) (BioLegend) and soluble anti-CD28 (2 μg/mL) (BioLegend) antibodies in RPMI (Merck), containing 10% Heat Inactivated FCS (Invitrogen) and 1% Penicillin-Streptomycin (Merck). Cell proliferation was measured as dilution of the cell dye as assessed by flow cytometry.

For in vitro polarization, naive T cells were freshly isolated and subsequently cultured for 4/5 days in RPMI (Merck), containing 10% Heat Inactivated FCS (Invitrogen) and 1% Penicillin-Streptomycin (Merck). Following conditions were used; Th1: 50 U/mL IL-2, 1 μg/mL anti-CD28, 1 μg/mL anti-CD3, 3.5 ng/mL IL-12, 10 μg/mL anti-IL4; Th2: 50 U/mL IL-2, 1 μg/mL anti-CD28, 1 μg/mL anti-CD3, 10 μg/mL IL-4, 10 μg/mL anti-IFN-γ; Th17: 1 μg/mL anti-CD28, 5 ng/mL TGFβ, 10 ng/mL IL-1b, 50 ng/mL IL-6, 20 ng/mL IL-23, 10 μg/mL anti-IFN-γ, 10 μg/mL anti-IL-4; Tregs: 50 U/mL IL-2, 1 μg/mL anti-CD28, 5 ng/mL TGFβ.

*Quantitative real-time PCR (qPCR)*. cDNA was synthesized from total RNA of isolated thymocytes and T cells, and qPCR performed according to the manufacturer's instruction (Bioline). All primers (Merck) were designed to span exon–exon boundaries and are available upon request. For the detection of Cct8 transcripts, primers spanning exon 2 was designed. Primer sequences are listed in Supplementary Data 4. The expression of the following genes was used to assess ER stress: Bcl2, Grp78, Atf6, Ire1a, Perk and Bim. Expression of Gapdh was used as an internal reference, and the delta Ct method was used in order to normalize expression. The delta delta CT (ΔΔCT) method was used for analyses of fold change, which was expressed as $2^{-\Delta\Delta CT}$.

*Phalloidin staining for nuclear actin filaments*. Phalloidin staining for nuclear actin was adapted from[14]. Briefly, CD4 T cells were isolated from lymph nodes of CCT8$^{T+/+}$ and CCT8$^{T-/-}$ mice at 4–8 weeks of age and plated O/N with 1 μg/mL anti-CD3 and 1 μg/mL anti-CD28 at 4 °C at a density of $10^5$ cells/mL in starvation media (RPMI (Merck), containing 0.5% Heat Inactivated FCS (Invitrogen) and 1% Penicillin-Streptomycin (Merck)) with wild-type cells added CK666 (Merck), an Arp2/3inhibitor, as negative control. The next day, cells were collected in 100 μL of starvation media, and allowed to adhere for 5 min on polyK-coated 35 mm glass-bottom dishes (Ibidi). Stimulation was performed by adding 100 μL of PMA/Iono solution (1:1000 cell stimulation cocktail (Invitrogen) in starvation media) drop-wise to the cell suspension for 30 s followed by permeabilization with a 100 μL mixture containing 0.3% Triton X-100+ Alexa Fluor Phalloidin 488 (1:2000) in cytoskeleton buffer (10 mM MES, 138 mM KCl, 3 mM MgCl, 2 mM EGTA and 0.32 M sucrose (pH 7.2)) for a maximum of 1 min. Cells were then fixed with 3 mL of 4% Formalin Solution (Merck) and incubated for 25 min. Fixed cells were washed twice with cytoskeleton buffer and stained with 1:500 30972 Abberior® STAR 635 (Merck) O/N at 4 °C. Stained cells were washed with cytoskeleton buffer

and imaged using a Leica SP8 STED microscope. STED was used for the detection of nuclear actin following a previously published method[14], thus allowing a direct comparison and allowing the imaging of sub-cellular structures in high resolution. To determine the number of cells with nuclear actin filaments, an average of 70 cells were scored for each of three independent experiments. For the cell images used in Fig. 3 the background was removed using the image processing software Fiji. Further, the cell membrane and nuclear envelope were removed by segmentation using the relevant function of the software.

*Proteomics.* 7575 naive CD4 cells were isolated from lymph nodes of CCT8[T+/+] and CCT8[T−/−] mice at 4–8 weeks of age by FACS sorting directly into lysis buffer (RIPA buffer (ThermoFisher Scientific) + 4% IGEPAL CA-360 (Merck)) for timepoint 0 h. At the same time, $10^6$ cells/mL FACS-sorted naive CD4 T cell were activated in vitro using a combination of plate-bound anti-CD3 (2 µg/mL) (BioLegend) and soluble anti-CD28 (2 µg/mL) (BioLegend) antibodies for 24 h before 7575 live cells were sorted directly into lysis buffer. Samples were stored at −20 °C until further processing. Samples were prepared for proteomic analysis using a modified SP3 method[65]. After thawing, 25 units of Benzonase (Merck) were added and samples incubated on ice for 30 min. Proteins were reduced with 5 mM dithiothreitol for 30 min at room temperature and then alkylated with 20 mM iodoacetamide for 30 min at room temperature. Then, 2 µL of carboxyl-modified paramagnetic beads were added to the samples (beads were prepared as in Hughes et al.)[66] along with acetonitrile to a concentration of 70% (v/v). To facilitate protein binding to the beads, samples were vortexed at 1000 rpm for 18 min. The beads were then immobilized on a magnet, the supernatant discarded and beads washed twice with 70% (v/v) ethanol and once with 100% acetonitrile. Washed beads were resuspended in 50 mM ammonium bicarbonate containing 25 ng trypsin (Promega) by brief bath sonication and incubated at 37 °C O/N. After digestion, the beads were resuspended by brief bath sonication and acetonitrile was added to 95% (v/v). Samples were vortexed at 1000 rpm for 18 min to bind peptides, then beads were immobilized on the magnet for 2 min and the supernatant discarded. Beads were resuspended in 2% DMSO, immobilized on the magnet for 5 min and the supernatant transferred to LC-MS vials which were stored at −20 °C until analysis.

Peptides were analysed by LC-MS/MS using a Dionex Ultimate 3000 UPLC coupled online to an Orbitrap Fusion Lumos mass spectrometer (ThermoFisher Scientific). A 75 µm × 500 mm C18 EASY-Spray column with 2 µm particles (ThermoFisher Scientific) was used with a flow rate of 250 nL/min. Peptides were separated with a linear gradient which increased from 2–35% buffer B over 60 min (A: 5% DMSO, 0.1% formic acid in water; B: 5% DMSO, 0.1% formic acid in acetonitrile). MS1 Precursor scans were performed in the Orbitrap at 120,000 resolution using an AGC target of 4e5 and a maximum cycle time of 1 s. Precursors were selected for MS/MS using an isolation window of 1.6 $m/z$ and were fragmented using HCD at a normalized collision energy setting of 28. Fragment spectra were acquired in the ion trap using the Rapid scan rate using an AGC target of 4e3.

Raw data files from each of the independent experiments were searched separately against the Uniprot mouse database (Retrieved 2017/03/15; 59094 entries) using MaxQuant[67] (version, 1.6.2.6). Trypsin specificity with two missed cleavages was specified, along with carbamidomethylation of cysteine as a fixed modification, and oxidation of methionine and protein N-terminal acetylation were allowed as variable modifications. The 'match between runs' option was enabled. Protein quantification was performed using the MaxLFQ[68] algorithm within MaxQuant and identifications were filtered to a 1% false discovery rate (FDR) on the peptide-spectral match and protein levels.

Differential analysis of proteomic data was carried out using Perseus[69] with an overall FDR calculated compared to a shuffled set of protein abundances. The minimum FDR between two independent replicates was reported in summary data (minFDR) with the effect size summarized as the difference between normalized label-free quantification (LFQ) abundances.

*Transcriptomics.* In all, 2000 naive CD4 T cells were isolated by FACS, sorting directly into lysis buffer (RLT buffer (Qiagen)) for timepoint 0 h. At the same time, FACS-sorted naive CD4 T cells ($10^6$ cells/mL) were activated in vitro using a combination of plate-bound anti-CD3 (2 µg/mL) (BioLegend) and soluble anti-CD28 (2 µg/mL) (BioLegend) antibodies for 24 h before 2000 live cells were FACS sorted directly into lysis buffer (RLT buffer (Qiagen)). RNA was extracted using the RNeasy Plus Micro kit (Qiagen). RNA was processed using the RNA-Seq Poly A method and 100 bp paired-end RNA-Seq was performed on the Illumina HiSeq4000 platform (Wellcome Trust Centre for Human Genetics, University of Oxford). Sequencing data were trimmed using Trimmomatic and aligned to the mm10 genome using STAR (version 2.5.3a)[70]. Reads were assigned to genes using HTSeq (intersection non-empty)[71]. Differential analysis was performed using edgeR[72]. Significant genes were defined as those with Benjamini-Hochberg adjusted $p$-values less than 0.05. ClusterProfiler was used for gene ontology (GO) analysis[73].

*Analysis of ER stress.* CCT8[T+/+] and CCT8[T−/−] naive CD4 cells were activated in vitro using a combination of plate-bound anti-CD3 (2 µg/mL) and soluble anti-CD28 (2 µg/mL) antibodies. As a positive control, CCT8[T+/+] cells were also cultured in the presence of 10 µg/mL Tunicamycin (Merck) to induce ER stress. After

24 and 48 h, viable cells were isolated by flow cytometry, RNA extracted (Qiagen) and cDNA synthesized (Bioline) for subsequent analysis by qPCR. Fold changes in gene expression were calculated relative to those of activated naive CCT8[T+/+] cells that had not been exposed to Tunicamycin.

*Metabolic assessments.* Splenic total CD4$^+$ T cells were isolated by magnetic sorting using commercial kit (Miltenyi Biotec) according to the manufacturer's protocol. Cells were plated at a density of $1 \times 10^6$ cells/mL and activated with Mouse T-Activator CD3/CD28 beads according to the manufacturer's manual (Thermo-Fisher Scientific) in RPMI (Merck), supplemented with 10% Heat Inactivated FCS (Invitrogen) and 1% Penicillin-Streptomycin (Merck) for 24 h. For all the following analyses, live cell stains were performed in situ followed by incubation in the dark, in a humidified, gassed (5% $CO_2$) incubator at 37 °C for 30 min. Active mitochondria were measured by adding 0.5 ng/mL Mitotracker DR (Invitrogen), while MitoID (Enzo Life Sciences) was added to measure total mitochondria, according to the manufacturer's protocol. Reactive oxygen species was measured by adding 5 µM MitoSox (ThermoFisher Scientific) to the cell cultures. Lipid droplets were analysed by the addition of 1 µg/mL Nile red, for measurement of palmitate uptake cells were incubated with 1 µg/mL BODIPY-FLC16 (ThermoFisher Scientific) and for uptake of glucose 5 µg/mL 6-NBDG (Invitrogen) was added. Apoptotic cells were detected using an Annexin-V/7AAD kit (BioLegend) with or without the addition of 1 µg/L N-Acetyl-L-cysteine (Merck). Mitochondrial membrane potential (Δψm) was measured with 2 µM JC-1 dye (ThermoFisher Scientific) by flow cytometry according to the manufacturer's directions. All staining experiments were performed at least 3 times with biological replicates.

*Measurement of metabolic flux.* Cellular metabolism was measured using an XF96 cellular flux analyser instrument from Seahorse Bioscience. OXPHOS was measured using a Mitostress test kit (Agilent Technologies) according to the manufacturer's instructions. Primary T cells, $3 \times 10^5$ per well, were cultured in RPMI with no sodium bicarbonate and 1% FCS, 20 mM glucose, 2 mM pyruvate and 50 µM β mercaptoethanol (pH 7.4) at 37 °C for these assays. Glycolysis was measured via ECAR measurements from the mitostress test data. Final drug concentrations used in the Seahorse assays were: Oligomycin 1 µM, FCCP 1.5 µM, Rotenone and Antimycin A both at 1 µM. Primary data were analysed using Wave desktop software from Agilent Technologies.

*Imaging flow cytometry.* Metabolic function was assessed by a 2 camera, 12 channel ImageStream X MkII (Amnis Corporation) with the 60x Multimag objective and the extended depth of field option providing a resolution of 0.3 µm per pixel and 16 µm depth of field. The side scatter laser was turned off to allow channel 6 to be used for PE-Cy7. Fluorescent excitation lasers and powers used were 405 nm (50 mW), 488 nm (100 mW) and 643 nm (100 mW). Bright field images were captured on channels 1 and 9 (automatic power setting). A minimum of 30,000 images were acquired per sample using INSPIRE 200 software (Amnis Corporation) and analysed by the IDEAS v 6.2 software (Amnis Corporation) using cells stained with single colour reagents, a colour compensation matrix was generated for all 10 fluorescence channels, run with the INSPIRE compensation settings and analysed with the IDEAS compensation wizard. Images were gated for focus (using the Gradient RMS feature) on both bright field channels (1 and 9) followed by selecting for singlet cells (DNA intensity/aspect ratio) and live cells at the time of staining, i.e. LIVE/DEAD aqua low-intensity (channel 8) or low-bright-field contrast (channel 1).

*Infection with* Heligmosomoides polygyrus. CCT8[T+/+] and CCT8[T−/−] mice were infected with 200 L3 *H. polygyrus* larvae by oral gavage. Adult parasites were eliminated by 100 mg/kg of pyrantel embonate (PyratapeP) by oral gavage. For secondary infection, drug-treated mice were challenged with another 200 L3 *H. polygyrus* larvae by oral gavage 10 days later. Faecal egg counts were used to assess the parasite burden and efficacy of the anti-helminthic treatment throughout the experiment and adult worm burdens were determined using standard procedures[74].

*IgG1 serum ELISA.* ELISA plates were coated with *H. polygyrus* excretory/secretory product (HES)[75] at 1 µg/mL in PBS O/N at 4 °C, washed 5 times with PBS containing 0.1% Tween (PBS-T) and then blocked using 2% BSA in PBS for 2 h at 37 °C. The plates were then washed a further 5 times in PBS-T, followed by mouse serum added at a 1:500,000 dilution in blocking solution in the first well with serial 3-fold dilutions across the ELISA plate and left to incubate in the fridge at 4 °C O/N. The following day the plates were washed with PBS-T and incubated with goat anti-mouse IgG1-HRP (Southern Biotech, USA) at 1:6000 dilution in blocking buffer and left for 1 h at 37 °C. After incubation, the plates were washed 5 times in PBS-T and a further 2 times in distilled water, after which ABTS substrate (KPL, USA) was added and the plates were read at 405 nm after 2.5 h, using PHERAstar FS plate reader (BMG LABTECH, Germany).

*Tuft cell staining.* Small intestines were isolated from mice shortly after euthanization and flushed with PBS, the gut was then inverted onto a wooden skewer and placed in 10% neutral buffered formalin (NBF, Merck) solution for 4 h. The tissue

was sliced longitudinally to remove from the skewer and tissue was then rolled up beginning at the ileum using the Swiss roll method as previously published[76]. The tissue was stored in NBF solution O/N then placed into 70% ethanol prior to paraffin embedding. Thin sections of paraffin-embedded tissue were de-waxed and stained with anti-DCAMLK1 at 1:1000 (Abcam, UK) and secondary stained using anti-rabbit-FITC, control slides were incubated with rabbit IgG (Abcam, UK). Once stained, slides were mounted in DAPI mounting media (Vector Laboratories, UK) and imaged using the 10x objective on a Leica DiM8 microscope (Leica, Germany); images were analysed using ImageJ/Fiji.

**Statistics and reproducibility**. Standard deviations and *p*-values were determined using GraphPad Prism software (Graph Pad Software Inc.). *p*-Values were calculated using a two-tailed unpaired Student's *t*-test with 95% confidence interval. Each figure legend indicates methods of comparison and corrections. All experiments have been carried out at least 2 times with biological triplicates in each group of the independent experiments, and no data were excluded from analysis. The experimental groups were determined by different genotype (i.e. wild-type versus homozygous CCT8-deficient) and were matched for general genetic background, age and gender. Control mice were non-Cre lox::lox animals, hence allowing the analysis of litter mates of individual matings with one parent heterozygous for the expression of Cre. The two experimental groups (i.e. wild-type versus homozygous CCT8-deficient) provided an obvious phenotype upon autopsy for the phenotypic analyses that blinding was not possible. In the infection experiments, blinding was not achieved due to the large worm burden in the CCT8-deficient mice infected.

**Reporting summary**. Further information on research design is available in the Nature Research Reporting Summary linked to this article.

## Data availability

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

## Acknowledgements

This work was supported by the Wellcome Trust through an Investigator Award to RMM (Ref. 106122), and the Wellcome Trust core-funded Wellcome Centre for Integrative Parasitology at the University of Glasgow (Ref: 104111). B.E.O. was supported by the Norwegian Research Council (Ref. 250030). S.D. was supported by a NDM studentship award (Oxford, UK).

## Author contributions

Conceptualization: G.A.H. and N.T.; methodology: G.A.H. and N.T; software: S.D. and A.H.; investigations: B.E.O., S.M., N.P., S.D., A.H., I.A.R., E.S., D.H., M.P.J.W., R.M.M. and M.E.D.; resources: G.A.H., D.H., R.M.M., R.F. and B.M.K.; data curating: A.H. and S.D.; writing: G.A.H, B.E.O., S.M. and A.H. with contributions from all.

## Competing interests

The authors declare no competing interests.
