## [Transparent Peer Review File. · Communications Biology]

Reviewers' comments:

Reviewer #1 (Remarks to the Author):

The manuscript by Oftedal et al presents data in which the CCT8 subunit of CCT chaperonin is knocked out in DP T cells, resulting in impaired thymic maturation and T cell homeostasis. Aberrant TH2 polarization is also described. The manuscript contributes timely information on the activities of CCT in supporting T cell-mediated immunity. The comments below suggest some directions to support the conclusions.

- 1) Only protein levels of CCT8 - one CCT subunit (of eight) - is evaluated) in Figure 1 . What are the protein levels of other CCT subunits (e.g. CCT2, CCT3 or CCT6) in thymic and T cells? Do these follow the same pattern of protein levels as CCT8. The authors infer that loss of a single subunit is enough to reduce the activity of the entire complex. While this is likely, it should be shown beyond the RNA data in Fig. 3bi. RNA levels may not correlate with protein and not all the CCT subunits were equally reduced (e.g. CCT2, CCT4).
- 2) Following the above comment – is there evidence that CCT8 could have activity as a monomer and/or independent of the complex and is this activity important for T cells?
- 3) Some discussion on the rationale for targeting CCT8 in preference to other subunits would be helpful. Also – loss of CCT is expected to inhibit cell proliferation and cause cell death – which could impact recovery of cells for analysis. However, this does not seem to be the case. Does loss of CCT8 alone have the same impact on T cell fitness as loss of all eight CCT subunits?
- 4) Expectations of loss of CCT protein folding activity would be at the protein level and indirectly at the gene level. Hence the gene expression data in Fig. 3bii is less compelling. Data shown in Fig. 3ci-ii is central to the conclusions made on the results of CCT8 loss in T cells but only one cell is shown in the image and quantitation is also from a few cells. Because of its importance, this body of work could be strengthened with additional experiments.
- 5) Increased IFN γ and TH1 in preference to TH2 is an intriguing finding – especially since the authors mention that they could not detect T-bet or IL-12R. This could be explored further such as showing additional data for upregulation of STAT1 and expanding upon the data mentioned in supplementary Fig 2. On this note – some clarity on how the data for Figure 3d was acquired would be helpful.
- 6) Minor point – some typos need correcting like line 214 (think you meant Figure 4a not 2a) and making sure that a figure is cited for all conclusions made.

Reviewer #2 (Remarks to the Author):

The manuscript by Oftedal et al. provides evidences in the mouse model for an important role of CCT in proteome changes induced by the activation. This is associated to a defective mature T cell activation and Th2 mediated immune responses against helminths.

The information is original and interesting for the field of immunology making first relationship between CCT and T cell immune responses. However, some experiments and stats should be better explained/done. It is also noticeable that there are more drastic differences in CD8 T cells, while experiments in the manuscript mainly focus on CD4 T cells. This referee thinks that some experiments to figure out the function of CD8 T cells would be helpful.

Particular comments:

1.- Is it possible to show the complete Western blot of figure 1a?

2.- T cell expansion protocol in figure 3a should be explained in results. More importantly the stats in survival experiments is not clear. Are the authors comparing data with IL-2 and without IL-2 (CCT+ vs CCT-)? This comparison should be done in order to claim the effect or not of IL-2. If IL-2 does not have an effect on survival, as the authors claim, the results in terms of stats should be similar.

Comparison should be accordingly described in the figure legend.

3.- In figure 3b the hit map show no differences of cct5 in 0h, at least are not easy to see. It is also not clear for this referee what stats are applied in this data. For example, are differences of cct7 statistically significant? Is it possible to support the loss of expression of at least some components by Western blot?

4.- Please describe in the text what stimulation is applied to the cells in figure 3b. It is also not clear for this referee why cells are stimulated in these assays. Maybe also explain that.

5.- In figure 3bii the same stats concerns applied. Why is different Tuba1b and not different Actb1 in 0 hours. I think this should be more carefully described/discussed and stats should support it and be clearer explained (in general in figure 3). Again some Western blots would assist in the interpretation of the data.

6.- Please explain why STED microscopy is required in figure 3c. It should be detailed how many cells were analysed per experiment. What does triplicate means in these experiments. More than 3 cells should be analysed. Were these assays done also with CD8 T cells?

7.- In figure 3d it would be also helpful to know what differences are statistically different, at least for those molecules that are mention in the text.

8.- In figure 4 it would be useful to explain in results section how cells where stimulated and the comparison of the results with and without tunicamycin.

9.- As mentioned above, this referee misses some information on CD8 T cell function. More important alterations are observed in CD8 cells, which do not express CCT as shown in figure 1a, and the majority of data are obtained with CD4 T cells. We have in the in vivo data on the function of eosinophils, B cells, ILCs, but there is no data on CD8 T cells. I am aware that this is not the most adequate in vivo model but in vitro experiments of activation, expansion, differentiation (as for CD4 T cells in figure 4) or immunological synapse assembly and secretory granule polarisation of CD8 T cells could be provided. This is important due to the role of actin in these processes, as the authors discuss in the second paragraph of the discussion.

10.- It would be maybe interesting some discussion on the role of chaperonine dysfunction in immunodeficiency, if any. Or maybe, does the CCT- phenotype in mice reminds any immunodeficiency? Do the authors think that this focus might be interesting for discussion of the manuscript?.

Minor comments

1.- May be please detail in the introduction what is known about the role of CCTs in T cell biology/function, if any.

2.- Also, is it any reason to select CCT8 instead of other subunit?

3.- Please correct typo in figure legend, line 780 `...data was...` and in terms of stats should read `...data were compared...` may be better than `...data was calculated...`. Also line 199 should be `...when unresolved...` instead of `...where unresolved...` (I guess) or in discussion there is a close repetition of taken together (lines 434 and 437). Thus, the text should be inspected for this kind of mistakes.

Response to reviewer's comments on the manuscript entitled "T cell maturation, selection, and function differentially depend on the chaperonin CCT" (COMMSBIO-20-2343-T) by B.E. Oftedal and colleagues.

Reviewer #1 (Remarks to the Author):

The manuscript by Oftedal et al presents data in which the CCT8 subunit of CCT chaperonin is knocked out in DP T cells, resulting in impaired thymic maturation and T cell homeostasis. Aberrant TH2 polarization is also described. The manuscript contributes timely information on the activities of CCT in supporting T cell-mediated immunity. The comments below suggest some directions to support the conclusions.

1) Only protein levels of CCT8 - one CCT subunit (of eight) - is evaluated in Figure 1. What are the protein levels of other CCT subunits (e.g. CCT2, CCT3 or CCT6) in thymic and T cells? Do these follow the same pattern of protein levels as CCT8. The authors infer that loss of a single subunit is enough to reduce the activity of the entire complex. While this is likely, it should be shown beyond the RNA data in Fig. 3bi. RNA levels may not correlate with protein and not all the CCT subunits were equally reduced (e.g. CCT2, CCT4).

We thank the reviewer for the comment and agree that RNA levels do not necessarily correlate with protein detection. While we have only verified the loss of the CCT8 subunit in thymic T cells for reasons now explained in the revised manuscript (lines 108-110, page 4), the results in Figure 3bi show protein expression data as measured by mass spectrometry. The results displayed in the figure therefore provide the information requested by the reviewer, namely protein levels of all CCT subunits. The Figure legend details this fact both in the title and in the information provided for panel 3b. Quantification of data in the form of Supplemental Table 2 (in addition to the heat map in panel bi of Figure 3) is now included showing the fold changes in protein expression comparing CCT8-deficient and -proficient CD4+ T cells (lines 167 and 176, page 6).

2) Following the above comment – is there evidence that CCT8 could have activity as a monomer and/or independent of the complex and is this activity important for T cells?

We appreciate the reviewer's interesting question. We have not probed the existence or function of monomeric CCT8 in T cells and have also not found in the literature a description of such structures occurring naturally in mammalian cells. However, yeast CCT8 monomers have been generated under experimental conditions and shown to remain ADP-bound and barely involved in the ATPase-cycle when compared to the complete CCT complexes (Jin et al., *An ensemble of cryo-EM structures of TRiC reveal its conformational landscape and subunit specificity*, PNAS, 2019). Based on this observation, it is very likely that monomeric CCT8 complexes, should they naturally exist, would have a minimal (if any) catalytic function.

3) Some discussion on the rationale for targeting CCT8 in preference to other subunits would be helpful. Also – loss of CCT is expected to inhibit cell proliferation and cause cell death – which could impact recovery of cells for analysis. However, this does not seem to be the case. Does loss of CCT8 alone have the same impact on T cell fitness as loss of all eight CCT subunits?

In response to the reviewer's helpful comment, we have now added in the revised version of our manuscript further reasoning why our work focused on the deletion of the θ subunit of CCT (line 108 - 110 on page 4; see also answer 1 to the reviewer's questions). In short, CCT8 is not only most strongly upregulated upon T cell activation as shown in Figure 3bi and independently reported elsewhere (P. Ojala et al. *Electrophoresis* 2007, 28, 903–917) but also plays a central role in controlling the intact TriC complex (Noormohammadi, A. et al. *Nat Commun* 7, 13649 (2016)).

We agree with the reviewer that a loss of CCT expression inhibits cell proliferation and causes cell death leading to early embryonic death (ref # 12 as well as Kim, A., Choi, K. *TRiC/CCT chaperonins are essential for organ growth by interacting with insulin/TOR signaling in Drosophila*. *Oncogene* 38, 4739–4754 (2019), for a complementary experimental system). We therefore adopted an experimental design whereby a loss of CCT8 expression was targeted exclusively to T cells. Upon *in vitro* activation, T cells deficient in CCT8 expression displayed a higher frequency of cell death (see revised Figure 3aii and the remaining viable cells displayed a lower proliferative index when compared to wild type controls (Figure 3ai). Moreover, the frequency and absolute number of mature thymocytes was only mildly reduced in CCT8T^{-/-} mice in comparison to wild type controls. In striking contrast, this difference was much larger for mature peripheral T cells, a finding in concordance with our *in vitro* results and further underscored by the *in vivo* inability of T cells to expand homeostatically (Figure 2b).

4) Expectations of loss of CCT protein folding activity would be at the protein level and indirectly at the gene level. Hence the gene expression data in Fig. 3bii is less compelling. Data shown in Fig. 3ci-ii is central to the conclusions made on the results of CCT8 loss in T cells but only one cell is shown in the image and quantitation

is also from a few cells. Because of its importance, this body of work could be strengthened with additional experiments.

We appreciate the reviewer's remarks but conclude that a misunderstanding may be the base to this question (see also question 1). Data presented in Figure 3bi + ii quantifies protein detected in resting and activated T cells, respectively, using mass spectrometry. The Figure legend details this fact both in the title and in the information provided for panels 3bi and ii.

Data shown in Figure 3ci-ii is representative of 3 independent experiments in which a total of 70 cell nuclei were inspected. This information, which had already been present in the Materials and Methods section of the paper initially submitted, has for clarity now also been added to the legend of Figure 3, line 799-800, page 23.

5) Increased IFN γ and TH1 in preference to TH2 is an intriguing finding – especially since the authors mention that they could not detect T-bet or IL-12R. This could be explored further such as showing additional data for upregulation of STAT1 and expanding upon the data mentioned in supplementary Fig 2. On this note – some clarity on how the data for Figure 3d was acquired would be helpful.

We welcome the reviewer's remark and the suggestion to probe activated CCT8-proficient and – deficient T cells for the detection of STAT1 and its state of activation. We noted, as shown in Figure 3d, that peripheral T cells from CCT8T^{-/-} mice had both before and after activation higher concentrations of STAT-1 proteins when compared to wild type T cells (ANOVA FDR = 0.02). The significance in the fold changes of proteins shown in Figure 3d are now also displayed in more detail in the Supplemental table 2 and mentioned in the revised text (line 176, page 6). We have added now also added the analysis of total STAT1 and pSTAT1 in resting and activated T cells to the revised manuscript (Figure 3 f), lines 202-204, page 7.

6) Minor point – some typos need correcting like line 214 (think you meant Figure 4a not 2a) and making sure that a figure is cited for all conclusions made.

We appreciate the reviewer's careful reading of the text and apologize for the typos. We have paid attention to correct these mistakes in the revised manuscript.

Reviewer #2 (Remarks to the Author):

The manuscript by Oftedal et al. provides evidences in the mouse model for an important role of CCT in proteome changes induced by the activation. This is associated to a defective mature T cell activation and Th2 mediated immune responses against helminths.

The information is original and interesting for the field of immunology making first relationship between CCT and T cell immune responses. However, some experiments and stats should be better explained/done. It is also noticeable that there are more drastic differences in CD8 T cells, while experiments in the manuscript mainly focus on CD4 T cells. This referee thinks that some experiments to figure out the function of CD8 T cells would be helpful.

Particular comments:

1.- Is it possible to show the complete Western blot of figure 1a?

In response to the reviewer's question we provide below one of the Western blot analyses which form the basis to Figure 1a. Please note that the filter was probed first with an anti-CCT8 antibody (left microphotograph) and then with an anti-GAPDH antibody (right microphotograph), without stripping the filter in between. As demonstrated the detection of GAPDH did not display additional bands that were not observed with the initial detection of CCT8.

2.- T cell expansion protocol in figure 3a should be explained in results. More importantly the stats in survival experiments is not clear. Are the authors comparing data with IL-2 and without IL-2 (CCT+ vs CCT-)? This comparison should be done in order to claim the effect or not of IL-2. If IL-2 does not have an effect on survival, as the authors claim, the results in terms of stats should be similar. Comparison should be accordingly described in the figure legend.

We thank the reviewer for highlighting the need to describe better how the analysis shown in Figure 3a was performed. In response to the reviewer's comment, we have amended Figure 3a_{ii} and added a Supplemental Table 1 to show the survival curve of the CCT8^{T+/+} and CCT8^{T-/-} T cells with and without supplementation of IL-2 and display appropriate statistics. We have also edited the manuscript accordingly (line 154-157, page 6).

3.- In figure 3b the hit map shows no differences of cct5 in 0h, at least are not easy to see. It is also not clear for this referee what stats are applied in this data. For example, are differences of cct7 statistically significant? Is it possible to support the loss of expression of at least some components by Western blot?

We note the reviewer's remarks and question related to Figure 3b. Both heat map panels of Figure 3b show a comparative analysis of proteomic data obtained by liquid-chromatography mass spectrometry (LC-MS/MS) to quantify Individual subunits of CCT and a set of typical targets (actins and tubulins). The profiles of protein expression for un-stimulated and activated T cells of either genotype were compared. For example, CCT5, which is lowly expressed in un-stimulated T cells irrespective of the genotype, was only upregulated in wild type but not in mutant T cells. As all CCT subunits were expressed at significantly lower concentrations in CCT8 -deficient T cells, we concluded "the engineered lack of CCT8 expression reduced the expression of all components of the CCT complex (Figure 3b_i)," (line 161 – 162, page 6). In the revised manuscript we have now added a Supplemental table 2 displaying the fold changes of the proteins reported in Figure 3b. (Please see also answer # 5)

Figure legend: Proteomic data obtained from naïve CD4 cells before (0 hrs) and 24 hrs after activation with anti-CD3 and anti-CD28. The protein abundance profiles are represented as LFQ (Label-free quantification) intensity calculated by normalizing the protein intensity globally across all samples.

4.- Please describe in the text what stimulation is applied to the cells in figure 3b. It is also not clear for this referee why cells are stimulated in these assays. Maybe also explain that.

We appreciate the reviewer's useful comment and have added new text in the Results section to explain how T cells were activated in the experiments shown in Figure 3 (line 154-157, page 6). This edit is in addition to text we have kept in the section Materials and Methods of the original manuscript. A rationale for why we investigated both resting and activated T cells is now provided in the revised manuscript (line 108-110, page 4).

5.- In figure 3bii the same stats concerns applied. Why is different Tuba1b and not different Actb1 in 0 hours. I think this should be more carefully described/discussed and stats should support it and be clearer explained (in general in figure 3). Again some Western blots would assist in the interpretation of the data.

We note the reviewer's comments and understand them also to be related to question 3. Differences in the detection of CCT substrates in resting T cells are to be expected, as both CCT dependency and folding efficiency will vary for individual molecules. In data shown in Figure 3bii we wished to interrogate whether the concentrations of known CCT substrates were reduced consequent to the deletion of CCT8. Hence we did not address the issue of "Why is different Tuba1b and not different Actb1 in 0 hours" but wished to detail changes in concentrations of protein that depend on CCT8 for their folding following induction of cell stress. Supplemental Table 2 now displays the statistical analysis of changes in protein concentrations observed in resting and activated T cells proficient or deficient for CCT8 expression. The data shown in Figure 3b + d and in Supplemental Table 2 were obtained by liquid-chromatography mass spectrometry (LC-MS/MS) to quantify differences in protein concentrations, a robust, accurate, and reproducible method (Aebersold *et al.*, *Mol Cell Proteomics*. 2013 Sep; 12(9):2381-2382). We therefore think that this method is more quantitative than a Western blot analysis.

6.- Please explain why STED microscopy is required in figure 3c. It should be detailed how many cells were analysed per experiment. What does triplicate means in these experiments. More than 3 cells should be analysed. Were these assays done also with CD8 T cells?

We appreciate the reviewer's comment and request for further information. STED microscopy was chosen as the most suitable method to visualize possible changes to the delicate lattice of actin filaments as a consequence of CCT8 depletion. Indeed, we required a method that could scan over a small region with high speed, good penetration of depth and super-high resolution which does not require subsequent computational image reconstruction. STED microscopy meets these criteria and is hence superior when compared to other conventional imaging methods such as confocal microscopy (see also reference 19 of the manuscript).

Data shown in Figure 3ci-ii is representative of 3 independent experiments in which a total of 70 cell nuclei were inspected. This information, which had already been present in the Materials and Methods section of the original manuscript submitted, has for clarity now also been added to the legend of Figure 3. Given the extent of functional deficiencies observed in CCT-deficient CD4+ T cells, we restricted this labour intensive analysis to this subpopulation.

7.- In figure 3d it would be also helpful to know what differences are statistically different, at least for those molecules that are mention in the text.

We concur with the reviewer's comments and have included in the revised manuscript the requested information as Supplemental table 2.

8.- In figure 4 it would be useful to explain in results section how cells where stimulated and the comparison of the results with and without tunicamycin.

In response to the reviewer's helpful comments, we have edited the corresponding paragraph in the Results section to detail how the cells were treated for the cell stress experiments shown in Figure 4 (line 220-225, page 7). This addition now complements information that has previously been provided in the section Materials and Methods.

To recapitulate, we generated a positive control for our experiments to probe ER stress and used for this purpose naïve wild type cells that had been activated for 48 hrs with both anti-CD3 and anti-CD28 antibodies and exposed to Tunicamycin. The fold difference for the CCT8T^{-/-} cells and the positive control are shown relative to the gene expression in the CCT8T^{+/+} cells arbitrarily set to a value of 1 (see hatched bars in Figure 4a). ER stress related transcripts were measured in this positive control

and activated CCT-deficient and -proficient T cells that were not exposed to the drug. This information is now added in the revised manuscript (line 648-649, page 19).

9.- As mentioned above, this referee misses some information on CD8 T cell function. More important alterations are observed in CD8 cells, which do not express CCT as shown in figure 1a, and the majority of data are obtained with CD4 T cells. We have in the *in vivo* data on the function of eosinophils, B cells, ILCs, but there is no data on CD8 T cells. I am aware that this is not the most adequate *in vivo* model but *in vitro* experiments of activation, expansion, differentiation (as for CD4 T cells in figure 4) or immunological synapse assembly and secretory granule polarisation of CD8 T cells could be provided. This is important due to the role of actin in these processes, as the authors discuss in the second paragraph of the discussion.

We appreciate the reviewer's comments and agree that CD8 T cells are also affected by the absence of CCT8, albeit to a significantly smaller extent. In addition to data shown in Figures 1 + 2, and in Supplemental Figures 1 + 2, we have added for the reviewer a selection of results derived from additional experiments carried out to probe specifically CD8 T cells (see below). In view of these results, we think a more detailed analysis of CD8 T cells is not warranted for this manuscript as they are expected to be by far less informative. In this context, we also wish to emphasize that the *H. polygyrus* helminth model was selected because immunity to this parasite is entirely CD4+ T cell-dependent and a depletion of CD8 T cells has no appreciable effect on either the immunological or parasitological parameters measured (Urban, J.F., Jr., Katona, I.M., and Finkelman, F.D. (1991) *Heligmosomoides polygyrus*: CD4+ but not CD8+ T cells regulate the IgE response and protective immunity in mice. *Experimental Parasitology* 73: 500-511). Hence, a different experimental *in vivo* system would be needed to probe adequately the behavior of CCT8-deficient CD8+ T cells. This is, however, in our view, well beyond the scope of this first manuscript on the role of CCT8 in T cell biology.

Figure legend: Probing thymocyte differentiation along the CD8 lineage and molecular analysis of the ER stress of CD8 T cells in response to cell activation. (a) Thymocyte negative selection of cortical CD8SP cells. (b) Thymocyte negative selection of medullary semimature (i) and mature (ii) CD8SP

cells. (c) Percentage of semi-mature CD69+MHC-I⁻ (SM), mature stage 1 CD69+MHC-1+ (M1) and mature stage 2 CD69-MHC-i+ (M2) thymocytes expressing cleaved caspase 3 as a biochemical marker for imminent programmed cell death. (d) TNF α production by CD8SP thymocytes isolated from M2 thymocytes activation by anti-CD3 and anti-CD28 for 48 hrs. (e) Molecular analysis of T cell activation mediated ER stress in CCT8-proficient and -deficient T cells 48 hrs. after activation with anti-CD3 and anti-CD28 antibodies. In panels a–d, grey bars denote CCT8-proficient and white bars CCT8-deficient cells. Left panels in a and b shown representative contour plots of the data shown in bar graphs and display the chosen gating strategy. *p<0.05, **p<0.01, ***p<0.001, ****p<0.0001 (Student's t test). Data shown in panel a, b, c, and d show the mean \pm SD are representative of 2 independent experiments, while panel e show one experiment with the average of three replicates for each gene.

10.- *It would be maybe interesting some discussion on the role of chaperonine dysfunction in immunodeficiency, if any. Or maybe, does the CCT- phenotype in mice reminds any immunodeficiency? Do the authors think that this focus might be interesting for discussion of the manuscript?*

We thank the reviewer for this interesting question. Germ-line deficiencies in CCT function are early embryonic lethal and correlate with the cells' increased death and limited proliferation. We would therefore not expect to observe a primary immunodeficiency in an individual with a loss of CCT8 (or for that matter CCT function altogether) in light of CCT8's essential role for assembling a complete CCT complex (see Figure 3bi). Indeed, the *Cct8* gene is not among the reported loci whose loss of function has been identified as a cause of a primary immunodeficiency. We therefore have omitted a speculative discussion of whether a spontaneous loss of CCT8 function in humans would result in a clinically observed congenital pathology.

Minor comments

1.- *May be please detail in the introduction what is known about the role of CCTs in T cell biology/function, if any.*

We appreciate the reviewer's comment and have expanded the Introduction to provide more information to the known and documented role of CCT in T cell biology (108-110, page 4)

2.- *Also, is it any reason to select CCT8 instead of other subunit?*

In response to the reviewer's helpful comment we have now provided in the revised manuscript the rationale why we specifically analyzed CCT8 and its depletion in T cells (line 108 – 110, page 4).

3.- *Please correct typo in figure legend, line 780 '...data was...' and in terms of stats should read '...data were compared...' 'may be better than '...data was calculated...'. Also line 199 should be '...when unresolved...' instead of '...where unresolved...' (I guess) or in discussion there is a close repetition of taken together (lines 434 and 437). Thus, the text should be inspected for this kind of mistakes.*

We appreciate the reviewer's careful reading of our manuscript and apologize for the typos and lack of precision and conciseness in a few instances. We have paid attention to omit these mistakes in the revised manuscript.

List of changes in the manuscript:

Line 21: Present address: Science for Life Laboratory, Department of Women's and Children's Health, Karolinska Institutet, Solna, Sweden" has been added to address 7.

Line 50-51: Rephrased a sentence in the abstract

Line 108-1010: We have added the requested information regarding choosing CCT8 for this study

Line 109: Reference added *P. Ojala et al. Electrophoresis 2007, 28, 903–917*

Line 110: Reference added *Noormohammadi, A. et al. Nat Commun 7, 13649 (2016).*

Line 154-157: Additional information about the activation of T cells have been added

Line 158: "was" corrected to "could not be"

Line 159: "nor" corrected to "or the"

Line 160: Reference to Supplementary Table 1 is added

Line 167: Reference to Supplemental Table 2 is added

Line 176: Reference to Supplemental Table 2 is added

Line 202-204: Added "Moreover, flow cytometry confirmed that STAT1 and its activated form, phospho-STAT1 (pSTAT1) were increased both before and 24 hours after mitogenic stimulation (Figure 3f)."

Line 220-224: The culturing condition for the ER stress experiments are added as requested

Line 224-225: "CCT8T^{-/-} and treated CCT8T^{+/+}" added for clarification

Line 229: Figure 2a is corrected to Figure 4a

Line 229-230: "These changes were comparable for both experimental groups albeit not of identical extent." is added for clarity
Line 246: "and pSTAT1" is added
Line 275: "an anthelmintic such as" is added for clarity
Line 316: Added reference to Supplemental Figure 5
Line 328-332: Rephrased for clarity
Line 386-389: STAT1 data discussed, and reference *Bocchini, et al. JAK-STAT 3, e970459 (2014)* added
Line 421: "and subsequent cytokine secretion" added for clarity
Line 423: "overtly" deleted
Line 453 "harmed" replaced by "impaired"
Line 454: "take together" changed to "In this study"
Line 460: TH1 changed to Th1
Line 501-505: Information about the STAT1 experiment is added to the methods
Line 551: "of three independent experiments" has been added for clarification
Line 572-573: As requested, the rationale for activating the T cells was added
Line 648-649: Information of the calculation of the gene expression of ER stress components has been added
Line 799-800: "an average of 70 nuclei inspected in three independent experiments" has been added as requested.
Line 802-804: Figure legend for figure 3f has been added
Line 817: "calculated" has been corrected to "compared"

Reviewers' comments:

Reviewer #1 (Remarks to the Author):

The authors satisfactorily addressed comments in the critique. There are no additional concerns.

Reviewer #2 (Remarks to the Author):

Particular comments:

1.- Is it possible to show the complete Western blot of figure 1a?

In response to the reviewer's question we provide below one of the Western blot analyses which form the basis to Figure 1a. Please note that the filter was probed first with an anti-CCT8 antibody (left microphotograph) and then with an anti-GAPDH antibody (right microphotograph), without stripping the filter in between. As demonstrated the detection of GAPDH did not display additional bands that were not observed with the initial detection of CCT8.

The WB looks actually nice. It's OK to me

2.- T cell expansion protocol in figure 3a should be explained in results. More importantly the stats in survival experiments is not clear. Are the authors comparing data with IL-2 and without IL-2 (CCT+ vs CCT-)? This comparison should be done in order to claim the effect or not of IL-2. If IL-2 does not have an effect on survival, as the authors claim, the results in terms of stats should be similar. Comparison should be accordingly described in the figure legend.

We thank the reviewer for highlighting the need to describe better how the analysis shown in Figure 3a was performed. In response to the reviewer's comment, we have amended Figure 3a and added a Supplemental Table 1 to show the survival curve of the CCT8T+/+ and CCT8T-/- T cells with and without supplementation of IL-2 and display appropriate statistics. We have also edited the manuscript accordingly (line 154-157, page 6).

Data looks nicer now, thank you for amendments. However, is it possible to add the statistical test done in sup table 1. The referee does not understand what does t-test with multiple comparisons means in the context of supp. Fig 2a. What was compared? I guess It was only compared CCT+ vs CCT-

Also in the text (line 157) it should be written that CD4 T cells were activated (not T cells generic) as results seems to be specific for CD4 T cells

3.- In figure 3b the hit map shows no differences of cct5 in 0h, at least are not easy to see. It is also not clear for this referee what stats are applied in this data. For example, are differences of cct7 statistically significant? Is it possible to support the loss of expression of at least some components by Western blot?

We note the reviewer's remarks and question related to Figure 3b. Both heat map panels of Figure 3b show a comparative analysis of proteomic data obtained by liquid-chromatography mass spectrometry (LC-MS/MS) to quantify individual subunits of CCT and a set of typical targets (actins and tubulins). The profiles of protein expression for un-stimulated and activated T cells of either genotype were compared. For example, CCT5, which is lowly expressed in un-stimulated T cells irrespective of the genotype, was only upregulated in wild type but not in mutant T cells. As all CCT subunits were expressed at significantly lower concentrations in CCT8 -deficient T cells, we concluded "the engineered lack of CCT8 expression reduced the expression of all components of the CCT complex (Figure 3bi)," (line 161 - 162, page 6). In the revised manuscript we have now added a Supplemental table 2 displaying the fold changes of the proteins reported in Figure 3b. (Please see also answer # 5)

I understand. However, I think it is not clear in the case of cct5 at 0 hours. The hit map looks the same. Also, when I am going to new supplementary table 2, I see that the difference in 0 hours for cct5 is -0,97, the same for cct6a, which looks different in the profile. I wouldn't question differences looking at the profiles, but in cct5 is actually not clear. Also, even these are data of proteomics, I think that a WB for any of the ccts will add to the confidence of the conclusion. Maybe on those with the highest differences. In my opinion, these data should be revised and complemented by other approach (WB, flow cytometry, under the microscope...).

In the graph authors send to me, I miss what is the statistical analysis. What text refers this figure to? Should I understand that these below are the same data than in figure 3bi? May be is pertinent to add this as a supplementary figure in the revised version of the manuscript (with all the information included that allow understanding it properly)

Figure legend: Proteomic data obtained from naïve CD4 cells before (0 hrs) and 24 hrs after activation with anti-CD3 and anti-CD28. The protein abundance profiles are represented as LFQ (Label-free quantification) intensity calculated by normalizing the protein intensity globally across all samples.

4.- Please describe in the text what stimulation is applied to the cells in figure 3b. It is also not clear for this referee why cells are stimulated in these assays. Maybe also explain that.

We appreciate the reviewer's useful comment and have added new text in the Results section to explain how T cells were activated in the experiments shown in Figure 3 (line 154-157, page 6). This edit is in addition to text we have kept in the section Materials and Methods of the original manuscript. A rationale for why we investigated both resting and activated T cells is now provided in the revised manuscript (line 108-110, page 4).

OK thanks for adding on that issue

5.- In figure 3bii the same stats concerns applied. Why is different Tuba1b and not different Actb1 in 0 hours. I think this should be more carefully described/discussed and stats should support it and be clearer explained (in general in figure 3). Again some Western blots would assist in the interpretation of the data.

We note the reviewer's comments and understand them also to be related to question 3. Differences in the detection of CCT substrates in resting T cells are to be expected, as both CCT dependency and folding efficiency will vary for individual molecules. In data shown in Figure 3bii we wished to interrogate whether the concentrations of known CCT substrates were reduced consequent to the deletion of CCT8. Hence we did not address the issue of "Why is different Tuba1b and not different Actb1 in 0 hours" but wished to detail changes in concentrations of protein that depend on CCT8 for their folding following induction of cell stress. Supplemental Table 2 now displays the statistical analysis of changes in protein concentrations observed in resting and activated T cells proficient or deficient for CCT8 expression. The data shown in Figure 3b + d and in Supplemental Table 2 were obtained by liquid-chromatography mass spectrometry (LC-MS/MS) to quantify differences in protein concentrations, a robust, accurate, and reproducible method (Aebbersold et al., Mol Cell Proteomics. 2013 Sep; 12(9):2381-2382). We therefore think that this method is more quantitative than a Western blot analysis.

Understand, authors are not interrogating the why. However, they claim that the detection of Actb1 was unaffected (line 168). Sorry, this was not clear to me looking at the profile. What I see is that the expression of this protein increases. Maybe the table helps. Obviously, if any of the proteins studied in figure 3bii decrease, these are tubulins.

It is clear that mass spectrometry is robust, I am aware of publications. However as mention in point 3 I think that complementary data should help. That's the opinion of this referee.

6.- Please explain why STED microscopy is required in figure 3c. It should be detailed how many cells were analysed per experiment. What does triplicate means in these experiments. More than 3 cells should be analysed. Were these assays done also with CD8 T cells?

We appreciate the reviewer's comment and request for further information. STED microscopy was chosen as the most suitable method to visualize possible changes to the delicate lattice of actin filaments as a consequence of CCT8 depletion. Indeed, we required a method that could scan over a small region with high speed, good penetration of depth and super-high resolution which does not require subsequent computational image reconstruction. STED microscopy meets these criteria and is hence superior when compared to other conventional imaging methods such as confocal microscopy (see also reference 19 of the manuscript).

Well, I do not see any microscopy data in reference 19. I do not have any problem with the STED. However, I do not think you need STED for nuclear actin detection and, as soon as you are not providing any information on the distribution, shape or density or any other spatial parameter. You do not need either high speed or good penetration to claim about amount of nuclear actin (I think). Anyway, please explain your reasons in the text of the manuscript (at least in material and methods)

Data shown in Figure 3ci-ii is representative of 3 independent experiments in which a total of 70 cell nuclei were inspected. This information, which had already been present in the Materials and Methods section of the original manuscript submitted, has for clarity now also been added to the legend of Figure 3. Given the extent of functional deficiencies observed in CCT-deficient CD4+ T cells, we restricted this labour intensive analysis to this subpopulation.

I think this is the place where this information (about the n of the experiments, the graphs and statistics) should be (please note that the figure legend in the first submission was not acceptable). Although, now it is better organized, there is still important missing information. Please explain what are the bars indicated in 3ci-ii. Authors must provide the standard error of the mean in order to see what are differences among the three experiments done. This is how it should be done (although I am aware that this would not modify the conclusion claimed by the authors). For example, in 3f authors claim to show mean +/- SD, so in 3c-ii it should be also written in figure legends. Please, revise all figure legends for possible lacking info.

7.- In figure 3d it would be also helpful to know what differences are statistically different, at least for those molecules that are mention in the text.

We concur with the reviewer's comments and have included in the revised manuscript the requested information as Supplemental table 2.

OK, with sup table 2 the text is more supported by figures than in the first submission. However, I do not see Atp5b in the table (probably Atp5j2? Different nomenclature? If so, Why?). Please care about this sort of inconsistencies to help the reader. What is minFDR? It is not explained anywhere in the text. It should be done in Material and methods section.

8.- In figure 4 it would be useful to explain in results section how cells where stimulated and the comparison of the results with and without tunicamycin.

In response to the reviewer's helpful comments, we have edited the corresponding paragraph in the Results section to detail how the cells were treated for the cell stress experiments shown in Figure 4 (line 220-225, page 7). This addition now complements information that has previously been provided in the section Materials and Methods.

To recapitulate, we generated a positive control for our experiments to probe ER stress and used for this purpose naïve wild type cells that had been activated for 48 hrs with both anti-CD3 and anti-CD28 antibodies and exposed to Tunicamycin. The fold difference for the CCT8T^{-/-} cells and the positive control are shown relative to the gene expression in the CCT8T^{+/+} cells arbitrarily set to a value of 1 (see hatched bars in Figure 4a). ER stress related transcripts were measured in this positive control and activated CCT-deficient and -proficient T cells that were not exposed to the drug. This information

is now added in the revised manuscript (line 648-649, page 19).

Well, I do not find this explanation in lines 648-649...(I guess I should read in 656-657)

I appreciate the effort but I am sorry to say that authors should consider to rewrite the paragraph (starting at line 221). It seems that authors are initially compared cells activated (somehow, not explained at this point) (figure 4a) and then (next...) do an activation with anti-CD3 and anti-CD28 in a second experiment. The point is that in the figure legend you read only the activation with anti-CD3 and anti-CD28. Maybe I am stupid, but sorry this is really confusing to me.

In the section of results should be clear what was done and authors must do a further effort to get it.

9.- As mentioned above, this referee misses some information on CD8 T cell function. More important alterations are observed in CD8 cells, which do not express CCT as shown in figure 1a, and the majority of data are obtained with CD4 T cells. We have in the in vivo data on the function of eosinophils, B cells, ILCs, but there is no data on CD8 T cells. I am aware that this is not the most adequate in vivo model but in vitro experiments of activation, expansion, differentiation (as for CD4 T cells in figure 4) or immunological synapse assembly and secretory granule polarisation of CD8 T cells could be provided. This is important due to the role of actin in these processes, as the authors discuss in the second paragraph of the discussion.

We appreciate the reviewer's comments and agree that CD8 T cells are also affected by the absence of CCT8, albeit to a significantly smaller extent. In addition to data shown in Figures 1 + 2, and in Supplemental Figures 1 + 2, we have added for the reviewer a selection of results derived from additional experiments carried out to probe specifically CD8 T cells (see below). In view of these results, we think a more detailed analysis of CD8 T cells is not warranted for this manuscript as they are expected to be by far less informative.

For me, in experiments shown in figure 1b and 2b is clear that CD8 cells are also affected. I wouldn't say less. Regarding the information in the provided figure, If we pay attention to panel e, it is clear that, although CD8 cells do not have CCT8 like in the case of CD4 cells, the induction of ER stress proteins is completely different.

Well that's why I thought that it is interesting to add experiments with CD8 cells. I find interesting that the result is different. I insist, authors should include some experiments with CD8 cells, get a conclusion and discuss it in the context of the manuscript. In my opinion there is no reason hampering this.

In this context, we also wish to emphasize that the *H. polygyrus* helminth model was selected because immunity to this parasite is entirely CD4+ T cell-dependent and a depletion of CD8 T cells has no appreciable effect on either the immunological or parasitological parameters measured (Urban, J.F., Jr., Katona, I.M., and Finkelman, F.D. (1991) *Heligmosomoides polygyrus*: CD4+ but not CD8+ T cells regulate the IgE response and protective immunity in mice. *Experimental Parasitology* 73: 500-511). Hence, a different experimental in vivo system would be needed to probe adequately the behavior of CCT8-deficient CD8+ T cells. This is, however, in our view, well beyond the scope of this first manuscript on the role of CCT8 in T cell biology.

As I said in the first revision, I am aware that this is not the most adequate in vivo model but in vitro experiments of activation, expansion, differentiation (as for CD4 T cells in figure 4) or immunological synapse assembly and secretory granule polarisation of CD8 T cells could be provided.

Then in next papers the authors may work on proper models for CD8 cells if any parameter in in vitro experiments is affected.

Figure legend: Probing thymocyte differentiation along the CD8 lineage and molecular analysis of the ER stress of CD8 T cells in response to cell activation. (a) Thymocyte negative selection of cortical CD8SP cells. (b) Thymocyte negative selection of medullary semimature (i) and mature (ii) CD8SP cells. (c) Percentage of semi-mature CD69+MHC-I- (SM), mature stage 1 CD69+MHC-1+ (M1) and mature stage 2 CD69-MHC-i+ (M2) thymocytes expressing cleaved caspase 3 as a biochemical marker

for imminent programmed cell death. (d) TNF α production by CD8SP thymocytes isolated from M2 thymocytes activation by anti-CD3 and anti-CD28 for 48 hrs. (e) Molecular analysis of T cell activation mediated ER stress in CCT8-proficient and -deficient T cells 48 hrs. after activation with anti-CD3 and anti-CD28 antibodies. In panels a–d, grey bars denote CCT8-proficient and white bars CCT8-deficient cells. Left panels in a and b shown representative contour plots of the data shown in bar graphs and display the chosen gating strategy. * $p < 0.05$, ** $p < 0.01$, *** $p < 0.001$, **** $p < 0.0001$ (Student's t test). Data shown in panel a, b, c, and d show the mean \pm SD are representative of 2 independent experiments, while panel e show one experiment with the average of three replicates for each gene.

10.- It would be maybe interesting some discussion on the role of chaperonine dysfunction in immunodeficiency, if any. Or maybe, does the CCT- phenotype in mice reminds any immunodeficiency? Do the authors think that this focus might be interesting for discussion of the manuscript?.

We thank the reviewer for this interesting question. Germ-line deficiencies in CCT function are early embryonic lethal and correlate with the cells' increased death and limited proliferation. We would therefore not expect to observe a primary immunodeficiency in an individual with a loss of CCT8 (or for that matter CCT function altogether) in light of CCT8's essential role for assembling a complete CCT complex (see Figure 3bi). Indeed, the Cct8 gene is not among the reported loci whose loss of function has been identified as a cause of a primary immunodeficiency. We therefore have omitted a speculative discussion of whether a spontaneous loss of CCT8 function in humans would result in a clinically observed congenital pathology.

Ok, thank you very much for the discussion, I agree with authors on this part

Minor comments

1.- May be please detail in the introduction what is known about the role of CCTs in T cell biology/function, if any.

We appreciate the reviewer's comment and have expanded the Introduction to provide more information to the known and documented role of CCT in T cell biology (108-110, page 4)

OK

2.- Also, is it any reason to select CCT8 instead of other subunit?

In response to the reviewer's helpful comment we have now provided in the revised manuscript the rationale why we specifically analyzed CCT8 and its depletion in T cells (line 108 – 110, page 4).

OK

3.- Please correct typo in figure legend, line 780 `...data was...` and in terms of stats should read `...data were compared...` may be better than `...data was calculated...`. Also line 199 should be `...when unresolved...` instead of `...where unresolved...` (I guess) or in discussion there is a close repetition of taken together (lines 434 and 437). Thus, the text should be inspected for this kind of mistakes.

We appreciate the reviewer's careful reading of our manuscript and apologize for the typos and lack of precision and conciseness in a few instances. We have paid attention to omit these mistakes in the revised manuscript.

List of changes in the manuscript:

Line 21: "Present address: Science for Life Laboratory, Department of Women's and Children's Health, Karolinska Institutet, Solna, Sweden" has been added to address 7.

Line 50-51: Rephrased a sentence in the abstract

Line 108-1010: We have added the requested information regarding choosing CCT8 for this study

Line 109: Reference added P. Ojala et al. Electrophoresis 2007, 28, 903–917

Line 110: Reference added Noormohammadi, A. et al. Nat Commun 7, 13649 (2016).

Line 154-157: Additional information about the activation of T cells have been added

Line 158: "was" corrected to "could not be"
Line 159: "nor" corrected to "or the"
Line 160: Reference to Supplementary Table 1 is added
Line 167: Reference to Supplemental Table 2 is added
Line 176: Reference to Supplemental Table 2 is added
Line 202-204: Added "Moreover, flow cytometry confirmed that STAT1 and its activated form, phospho-STAT1 (pSTAT1) were increased both before and 24 hours after mitogenic stimulation (Figure 3f)."
Line 220-224: The culturing condition for the ER stress experiments are added as requested
Line 224-225: "CCT8T-/- and treated CCT8T+/+" added for clarification
Line 229: Figure 2a is corrected to Figure 4a
Line 229-230: "These changes were comparable for both experimental groups albeit not of identical extent." is added for clarity
Line 246: "and pSTAT1" is added
Line 275: "an anthelmintic such as" is added for clarity
Line 316: Added reference to Supplemental Figure 5
Line 328-332: Rephrased for clarity
Line 386-389: STAT1 data discussed, and reference Bocchini, et al. JAK-STAT 3, e970459 (2014) added
Line 421: "and subsequent cytokine secretion" added for clarity
Line 423: "overtly" deleted
Line 453 "harmed" replaced by "impaired"
Line 454: "take together" changed to "In this study"
Line 460: TH1 changed to Th1
Line 501-505: Information about the STAT1 experiment is added to the methods
Line 551: "of three independent experiments" has been added for clarification
Line 572-573: As requested, the rationale for activating the T cells was added
Line 648-649: Information of the calculation of the gene expression of ER stress components has been added
Line 799-800: "an average of 70 nuclei inspected in three independent experiments" has been added as requested.
Line 802-804: Figure legend for figure 3f has been added
Line 817: "calculated" has been corrected to "compared"

OK, thank you very much for the effort

Response to reviewers 2 comments:

Comment 1: This has been satisfactorily answered

Comment 2: The reviewer asks “Is it possible to add the statistical test done in sup table 1. The referee does not understand what does t-test with multiple comparisons means in the context of supp. Fig 2a. What was compared? I guess It was only compared CCT+ vs CCT-. Also in the text (line 157) it should be written that CD4 T cells were activated (not T cells generic) as results seems to be specific for CD4 T cells«

We used an unpaired Student's t test with adjustment for multiple comparisons to analyse the data displayed in Supplemental table 1. This detail has now been provided in Supplemental table 1. The data are parametric, and we adjusted for multiple comparisons using Bonferroni correction as we compared NAC - vs. NAC + for CCT+ and CCT-. We have corrected the text to specify that CD4+ cells were activated (line 157 of the manuscript).

Comment 3: The reviewer states: “I think it is not clear in the case of cct5 at 0 hours. The hit map looks the same. Also, when I am going to new supplementary table 2, I see that the difference in 0 hours for cct5 is -0,97, the same for cct6a, which looks different in the profile. I wouldn't question differences looking at the profiles, but in cct5 is actually not clear.

The heat maps in figure 3 are representative of one out of two replicate experiments as stated in the Figure legend. Because the proteomic analysis was independently performed on two separate occasions but provided comparable results, supplemental table 2 displays the average values of both sets of analyses. The heat map displays therefore a representative data set from one of the two experiments whereas the Table informs of the combined results of these studies.

«Also, even these are data of proteomics, I think that a WB for any of the ccts will add to the confidence of the conclusion. Maybe on those with the highest differences. In my opinion, these data should be revised and complemented by other approach (WB, flow cytometry, under the microscope...).»

The reviewer apparently favors the use of Western blotting to quantify differences in detected proteins in this particular instance differences between wild type and CCT8-deficient naive T cells. While we show this data for the protein that has been deleted by conditional gene targeting (as a verification of the mouse models fidelity) we have used the MS/MS to probe the proteome in an “unsupervised” fashion. We respectfully disagree with the reviewer's comment and specifically dispute the notion that “WB for any of the ccts will add to the confidence of the conclusion....and these data should be revised and complemented by other approach (WB, flow cytometry, under the microscope...)”. Western blotting and microscopy are not sufficiently quantitative to replace MS/MS and are in too many ways biased by the availability, quality and affinity of the antibodies used to detect differences in the expression levels of different proteins. Moreover, Western blotting is not really orthogonal and the dynamic range of protein detection by MS/MS is ~2 orders of magnitude larger when comparing to Western blots. Indeed, excellent sequence coverage for CCT proteins was observed in our experiments with 20-30 unique peptides per protein identified and quantified across the sample set allowing for a robust quantitation superior to a Western Blot with its single readout of low dynamic range. Hence, the suggestion to use Western blotting is therefore in our understanding neither helpful nor necessary ([https://www.mcponline.org/article/S1535-9476\(20\)31024-0/fulltext](https://www.mcponline.org/article/S1535-9476(20)31024-0/fulltext) and <https://www.nature.com/articles/nprot.2011.333>). Finally, we are certain that the use of MS/MS provides quantitative data sufficient for the statements made in our manuscript as LC-MS/MS is a robust, accurate, and reproducible to quantify differences in protein concentrations (see also Aebersold et al., Mol Cell Proteomics. 2013 Sep; 12(9):2381-2382). Finally, we contest the concept that quantification by flow cytometry should serve as an alternative to LC-MS/MS. FACS analysis requires intracytoplasmic staining with high-quality antibodies that may or may not recognize cryptic epitopes which are only accessible in denatured proteins and each of these antibodies will have with different affinities and generate separate individual background signals, thus rendering a comparative quantification not only challenging but merely indirect. In summary, we therefore conclude that additional Western blots, microscopy or FACS analyses are unnecessary to confirm our findings by LC-MS/MS and would also not be inline with good experimental practice and animal welfare to use only the absolutely necessary number of animals for experimental work.

In the graph authors send to me, I miss what is the statistical analysis. What text refers this figure to? Should I understand that these below are the same data than in figure 3bi? May be is pertinent to add this as a supplementary figure in the revised version of the manuscript (with all the information included that allow understanding it properly)»

The data provided in form of a bar graph and as part of our response to the reviewer's original comment is from the same analysis the data as in figure 3bi, and was added to illustrate that we do detect all CCT subunits.

Comment 4: these queries have been satisfactorily answered.

Comment 5: There reviewer writes “*Understand, authors are not interrogating the why. However, they claim that the detection of Actb1 was unaffected (line 168). Sorry, this was not clear to me looking at the profile. What I see is that the expression of this protein increases. Maybe the table helps. Obviously, if any of the proteins studied in figure 3bii decrease, these are tubulins. It is clear that mass spectrometry is robust, I am aware of publications. However as mention in point 3 I think that complementary data should help. That’s the opinion of this referee.*»

The profile from the heatmap reflects minor variations and none of these reached statistical significance. This robust conclusion is further documented in the supplementary table. As to the reviewer’s opinion to confirm the LC-MS/MS data sets with Western blotting we refer to our detailed answer given under point 3.

Comment 6: There reviewer notes “*Well, I do not see any microscopy data in reference 19. I do not have any problem with the STED*».

We thank the reviewer for this remark and apologize for the oversight and now refer to reference 14 in the revised manuscript. In addition, we have added the following text to the section Materials and Methods: “STED was used for the detection of nuclear actin following the previously published method¹⁴, thus allowing direct comparison and allowing the imaging of sub-cellular structures in high resolution.”

«*However, I do not think you need STED for nuclear actin detection and, as soon as you are not providing any information on the distribution, shape or density or any other spatial parameter. You do not need either high speed or good penetration to claim about amount of nuclear actin (I think). Anyway, please explain your reasons in the text of the manuscript (at least in material and methods)*»

STED was used for the detection of nuclear actin filaments after T cell activation since this was the method used in the Science immunology paper referenced in our manuscript (DOI: 10.1126/sciimmunol.aav1987) thus allowing direct comparison. STED is typically used in such analyses as this method allows the imaging of sub-cellular structures in higher resolution compared to other conventional imaging techniques like confocal microscopy. Of note, this analysis clearly provides the visual information of the shape and organization of nuclear F-actin. We therefore respectfully disagree with the reviewer’s opinion, conclusion and suggestion. Further microscopic analyses would indeed draw the same conclusions – if at all – what we have shown with STED but with significantly inferior data quality.

«*I think this is the place [referring to of Figure 3] where this information (about the n of the experiments, the graphs and statistics) should be (please note that the figure legend in the first submission was not acceptable). Although, now it is better organized, there is still important missing information. Please explain what are the bars indicated in 3ci-ii. Authors must provide the standard error of the mean in order to see what are differences among the three experiments done. This is how it should be done (although I am aware that this would not modify the conclusion claimed by the authors). For example, in 3f authors claim to show mean +/- SD, so in 3c-ii it should be also written in figure legends. Please, revise all figure legends for possible lacking info.*»

At the end of each figure legend the manuscript clearly stated which statistical test were used to calculate the data displayed. For Figure 3, we specifically wrote “Data shown in bar graphs represent mean \pm SD values of a single experiment representative of two independent experiments with three replicates each” in figure 3. Hence, this also applies to figure 3c ii, showing the mean \pm SD.

Comment 7: The reviewer states “*...I do not see Atp5b in the table (probably Atp5j2? Different nomenclature? If so, Why?). Please care about this sort of inconsistencies to help the reader. What is minFDR? It is not explained anywhere in the text. It should be done in Material and methods section.*»

We thank the reviewer for spotting this oversight. Indeed, the manuscript should have stated «Atp5j2». We have now updated the manuscript accordingly.

What is minFDR? It is not explained anywhere in the text. It should be done in Material and methods section.»

Differential analysis of proteomic data was carried out using Perseus with an overall false discovery rate (FDR) calculated compared to a shuffled set of protein abundances (https://link.springer.com/protocol/10.1007/978-1-4939-7493-1_7). The minimum FDR between two independent replicates was reported in summary data (minFDR) with the effect size summarised as the difference between normalised label-free quantification (LFQ)

abundances. This information has now been added to the material and methods, and the corresponding reference as has been cited as reference #69.

Comment 8: The reviewer remarks “Well, I do not find this explanation in lines 648-649...(I guess I should read in 656-657).”

In response to the reviewer’s remark, an explanation is now provided in the revised manuscript (line 657-658).

«I appreciate the effort but I am sorry to say that authors should consider to rewrite the paragraph (starting at line 221). It seems that authors are initially compared cells activated (somehow, not explained at this point) (figure 4a) and then (next...) do an activation with anti-CD3 and anti-CD28 in a second experiment. The point is that in the figure legend you read only the activation with anti-CD3 and anti-CD28. Maybe I am stupid, but sorry this is really confusing to me. In the section of results should be clear what was done and authors must do a further effort to get it.»

We have rephrased the paragraph as suggested by the reviewer (see line 219) and trust he/she finds the edits satisfactory. In addition, data related to ER stress in CD8+ cell is now included in this paragraph and shown in the revised Figure 4a.

Comment 9: The reviewer states: «For me, in experiments shown in figure 1b and 2b is clear that CD8 cells are also affected. I wouldn’t say less. Regarding the information in the provided figure, If we pay attention to panel e, it is clear that, although CD8 cells do not have CCT8 like in the case of CD4 cells, the induction of ER stress proteins is completely different. Well that’s why I thought that it is interesting to add experiments with CD8 cells. I find interesting that the result is different. I insist, authors should include some experiments with CD8 cells, get a conclusion and discuss it in the context of the manuscript. In my opinion there is no reason hampering this.»

In response to the reviewer’s comment, we have now expanded the manuscript demonstrating data related to (i) the spontaneous ER stress by CCT8-deficient CD8+ peripheral T cells (Figure 4a) and (ii) TNF- α -response of mature thymocytes (Figure 1g (ii)).

«As I said in the first revision, I am aware that this is not the most adequate in vivo model but in vitro experiments of activation, expansion, differentiation (as for CD4 T cells in figure 4) or immunological synapse assembly and secretory granule polarisation of CD8 T cells could be provided. Then in next papers the authors may work on proper models for CD8 cells if any parameter in in vitro experiments is affected»

We appreciate the reviewer’s suggestion how to design a next series of experiments for a follow-up report. This manuscript provides already significant information related to the *in vitro* and *in vivo* biology of CD8+ T cells devoid of CCT8 expression. Specifically, *in vivo* data is shown related to CD8 T cell functions (thymocyte selection, post-selectional maturation and function, peripheral T cell expansion and differentiation into distinct post-antigen exposure phenotypes). Moreover, we have now provided a detailed molecular analysis of the stress response of CD8 T cells. We are therefore of the firm understanding that a selection of additional *in vitro* analyses such as mitogenic activation, immunological synapse assembly and/or secretory granule polarisation of CCT8-deficient CD8 T cells are neither necessary nor essential to support our findings reported here. These efforts are clearly beyond the scope of the work described here. Last but not least, the requested experiments are technically very demanding to perform as the mice are severely lymphopenic and consequently there is a striking paucity of peripheral CD8+ T cells.

Comment 10: these queries have been satisfactorily answered.

List of new changes to the manuscript; all new changes are written in blue.

Supplemental table 1 Statistical analysis have been added
Supplemental table 2 "the table show the average difference across the two independent experiments" has been added.

Manuscript:
Line 155 "CD4+" have been added.
Line 182 Correcting ATP synthase subunit to ATP synthase membrane subunit f (Afp5j2) and correcting reference 24.
Line 219-233 This paragraph has been rewritten for clarity, and we have made a distinction between CD4 and CD8 cells.

Line 581-583 "STED was used for the detection of nuclear actin following the previously published method 14, thus allowing direct comparison and allowing the imaging of sub-cellular structures in high resolution. " has been added

Line 633-637 Inserted "Differential analysis of proteomic data was carried out using Perseus69 with an overall false discovery rate (FDR) calculated compared to a shuffled set of protein abundances. The minimum FDR between two independent replicates was reported in summary data (minFDR) with the effect size summarised as the difference between normalised label-free quantification (LFQ) abundances" to explain the minFDR

Line 780-781 "and SPCD8 cells" has been added to the figure legend of figure 1g

Line 816 Specification of the statistical method has been added

Line 824, 825 CD4 and CD8 added to the figure legend of figure 4a

Line 834-835 Specification of the statistical method has been added

Line 849 Specification of the statistical method has been added

Line 934 Exchanged reference 24

Line 1094 Added reference 69

Figures:

Figure 1g(ii) The TNF α production by CD8 cells has been added and the figure legend updated.

Figure 4a CD8 data is added for the ER stress response and the figure legend updated.

REVIEWERS' COMMENTS:

Reviewer #2 (Remarks to the Author):

The manuscript improved after modifications included by the authors. Still think that it would have been nice to have some experiments interrogating important processes dependent on correct actin dynamics, including those even mention by authors in their discussion

'.....Productive folding and processing of actin in T cells is an important prerequisite for the 359 signaling-dependent changes in shape. Indeed, T cells adapt to different physiological 360 conditions including, for example, the stress of the bloodstream flow, the migration to and 361 residence in different tissues each with their bespoke microenvironments, the engagement 362 with antigen-presenting cells and the formation of immunological synapses49...'

However, authors think this is out of scope of the manuscript. I accept it.

I find interesting the different results obtained with CD8 T cells when cellular stress is interrogated. However, i do not find any discussion about this in the discussion section. Please discuss why the role of CCT8 in CD8 cells is apparently different? This is my only pending issue.

Response to reviewers 2 comments:

Comment 1: *The manuscript improved after modifications included by the authors. Still think that it would have been nice to have some experiments interrogating important processes dependent on correct actin dynamics, including those even mention by authors in their discussion '.....Productive folding and processing of actin in T cells is an important prerequisite for the 359 signaling-dependent changes in shape. Indeed, T cells adapt to different physiological 360 conditions including, for example, the stress of the bloodstream flow, the migration to and 361 residence in different tissues each with their bespoke microenvironments, the engagement 362 with antigen-presenting cells and the formation of immunological synapses49...'*

However, authors think this is out of scope of the manuscript. I accept it.

Thank you for acknowledging the changes made to the manuscript. We look forward to follow-up our work with new studies.

Comment 2: *I find interesting the different results obtained with CD8 T cells when cellular stress is interrogated. However, i do not find any discussion about this in the discussion section. Please discuss why the role of CCT8 in CD8 cells is apparently different? This is my only pending issue.*

As the reviewer suggests, we have modified the text in the discussion, line 376-380 (the underlined text has been added):

Indeed, both ER stress and UPR play an important role in the homeostatic maintenance of peripheral T cells and are disordered in the absence of CCT, especially CD4⁺ T cells. In contrast, the UPR response of CD8⁺ T cells was minimally affected by the absence of CCT8 for reasons yet to be determined.

According to editorial instructions, the additional changes have been made:

Line 121: Added "Supplementary Data 1 and Supplementary Data 2"

Line 126: Replaced Figure 1bi with Figure 1b

Line 126: Replaced Figure 1bii with Figure 1c

Line 128: Replaced Figure 1c with Figure 1d

Line 129: Replaced Figure 1d with Figure 1e

Line 132: Replaced Figure 1d with Figure 1f and 1g

Line 134: Replaced Figure 1f with Figure 1h and 1i

Line 136: Replaced Figure 1bii and 1g with Figure 1c, 1j 1k

Line 138: Replaced Figure 1h with Figure 1l and 1m

Line 142: Added "Supplementary Data 1 and Supplementary Data 2"

Line 149 Added "and 2f"

Line 150: Replaced Figure 2f with Figure 2g

Line 162: Added “Supplementary Data 1 and Supplementary Data 2”

Line 169: Replaced “Figure 3bii and Supplementary Table 2” with “Figure 3c and Supplementary Data 3”

Line 173: Replaced Figure 3c with Figure 3d and 3e

Line 179: Replaced “Figure 3d and Supplementary Table 2” with “Figure 3f and Supplementary Data 3”

Line 184: Replaced Figure 3d with Figure 3f

Line 192: Replaced Figure 3e with Figure 3g and 3h

Line 193: Replaced Figure 3e with Figure 3g and 3h

Line 200: Replaced Figure 3c with Figure 3d

Line 207: Replaced Figure 3f with Figure 3i

Line 230: Added Supplementary Data 1

Line 245: Replaced Figure 4bi with Figure 4b, and added “Supplementary Data 1 and Supplementary Data 2”

Line 247: Replaced Figure 4bii with Figure 4c

Line 250: Replaced Figure 3d with Figure 3e

Line 253: Replaced Figure 4biii with Figure 4d

Line 255 Replaced Figure 4biv with Figure 4e

Line 266 Replaced Figure 4di with Figure 4f

Line 268 Replaced figure 4dii with Figure 4g

Line 269: Replaced Figure 4diii with Figure 4h

Line 289: Added “Supplementary Data 1 and Supplementary Data 2”

Line 291: Added -d

Line 298: Replaced Figure 5d with 5e

Line 300: Replaced Figure 5d with 5f

Line 311: Replaced Figure 5e with 5g

Line 319: Replaced Figure 5gi with Figure 5h

Line 321: Replaced Figure 5gii with Figure 5i

Line 323: Replaced Figure 5h with Figure 5j

Line 333 Replaced Figure 5h with Figure 5 k and l

Line 492: Replaced Key resource table with Supplementary Data 4

Line 546: Added The blot is shown as Supplementary Data 5.

Line 566: Replaced Key resource table with Supplementary Data 4

Line 745: “Quantification and Statistical Analysis” have been corrected to “Quantification, Statistical Analysis, and Reproducibility”, and the following text has been added to the section:

“All experiments have been carried out at least two times with biological triplicates in each group of the independent experiments, and no data was excluded from analysis. The experimental groups were determined by different genotype (i.e. wild type versus homozygous CCT8-deficient) and were matched for general genetic background, age and gender. Control mice were non-Cre lox::lox animals, hence allowing the analysis of litter mates of individual matings with one parent heterozygous for the expression of Cre. The two experimental groups (i.e. wild type versus homozygous CCT8-deficient) provided an obvious phenotype upon autopsy for the phenotypic analyses that blinding was not possible. In the infection experiments, blinding was not achieved due to the large worm burden in the CCT8-deficient mice infected.”

Line 785- 867: The Figure legends have been updated according to the changes made in the text.